# Learning from Imperfect Human Feedback: A Tale from Corruption-Robust Dueling

**Yuwei Cheng**
Department of Statistics
University of Chicago
Chicago, IL 60637, USA
yuweicheng@uchicago.edu

**Fan Yao**[*]
Department of Computer Science
University of Virginia
Charlottesville, VA 22904, USA
fy4bc@virginia.edu

**Xuefeng Liu**[*]
Department of Computer Science
University of Chicago
Chicago, IL 60637, USA
xuefeng@uchicago.edu

**Haifeng Xu**
Department of Computer Science
University of Chicago
Chicago, IL 60637, USA
haifengxu@uchicago.edu

## Abstract

This paper studies Learning from Imperfect Human Feedback (LIHF), addressing the potential irrationality or imperfect perception when learning from comparative human feedback. Building on evidences that human's imperfection decays over time (i.e., humans learn to improve), we cast this problem as a concave-utility continuous-action dueling bandit but under a restricted form of corruption: i.e., the corruption scale is decaying over time as $t^{\rho-1}$ for some "imperfection rate" $\rho \in [0, 1]$. With $T$ as the total number of iterations, we establish a regret lower bound of $\Omega(\max\{\sqrt{T}, T^\rho\})$ for LIHF, even when $\rho$ is known. For the same setting, we develop the Robustified Stochastic Mirror Descent for Imperfect Dueling (RoSMID) algorithm, which achieves nearly optimal regret $\tilde{\mathcal{O}}(\max\{\sqrt{T}, T^\rho\})$. Core to our analysis is a novel framework for analyzing gradient-based algorithms for dueling bandit under corruption, and we demonstrate its general applicability by showing how this framework can be easily applied to obtain corruption-robust guarantees for other popular gradient-based dueling bandit algorithms. Our theoretical results are validated by extensive experiments.

## 1 Introduction

Many real-world problems, such as personalized recommendation (Yue & Joachims, 2009; Immorlica et al., 2020; Yao et al., 2022) and fine-tuning generative models (Bai et al., 2022; Casper et al., 2023; Han et al., 2024), require learning from human feedback. Expressing preferences as numerical values or concrete functions is generally challenging for humans. Therefore, an approach that has achieved significant real-world success is to learn humans' preferences by eliciting their comparative feedback between two options (Bai et al., 2022). A natural theoretical framework capturing such learning task is the seminal dueling bandits framework introduced by Yue & Joachims (2009). The dueling bandit problem features a sequential online learning problem, during which a pair of actions are selected at each round and only their comparison result is revealed to the learner. The comparison outcome between two actions is modeled using a utility function of actions, alongside a link function that determines the probability of each action winning based on their utility difference. This modeling approach is termed as utility-based dueling bandit and achieves great success in generating summaries closely aligned with human preferences (Stiennon et al., 2020). However, human feedback is often *imperfect*, a crucial factor that has been largely overlooked in previous studies of dueling bandit. As we are all aware of, humans are not always rational (Posner, 1997) neither perfectly know our preferences (Pu et al., 2012). A powerful framework that can capture this problem is robust bandit learning under adversarial corruption (Bogunovic et al., 2020; Saha &

---

[*]Equal contribution.

Gaillard, 2022; Di et al., 2024). However, assuming fully adversarial feedback from a regular human seems over-pessimistic at the first glance, hence might make our algorithm development overly conservative. Indeed, ample behavioral studies show that humans often refine their preferences through interactions with the system, navigating a complex tradeoff between exploration and exploitation (Cohen et al., 2007; Wilson et al., 2014). Furthermore, their response errors may exhibit systematic patterns that depend on their past consumption history, rather than merely random noise. Though the sources of imperfections vary a lot in human feedback, one common finding in these studies is that *the feedback's imperfection tends to decrease over time*, as humans interact more with the system (Cohen et al., 2007; Wilson et al., 2014; Immorlica et al., 2020; Yao et al., 2022). This premise is the core motivation of this work, which studies dueling bandit learning under corruption that can be arbitrary but have decaying scales.

**Our Model and Contributions.** We cast Learning from Imperfect Human Feedback (LIHF) as a concave-utility continuous-action dueling bandit learning problem under adversarial utility corruption yet with decaying scale, upper bounded by $\mathcal{O}(t^{\rho-1})$ at round $t$, for some $\rho \in [0, 1]$ (hence the total amount of attack within time $T$ is $\mathcal{O}(T^{\rho})$). A fundamental research question that motivates our study is whether LIHF is fundamentally easier than learning under arbitrarily corrupted feedback.

Our first main technical result hints on a potentially negative answer to the above question. We prove a strong regret lower bound of $\Omega(d \max\{\sqrt{T}, T^{\rho}\})$ under LIHF, where $d$ is the action's dimension. This bound holds even when the decaying rate of imperfection index $\rho$ is known. This lower bound has the same order as recent lower bounds of dueling bandits under (the harder) arbitrary corruption (Agarwal et al., 2021; Di et al., 2024), thus hinting that learning from imperfect human feedback may be no easier than learning from arbitrary adversarial corruption. Notably, however, we use the phrase "hint", while not an affirmative claim, because our setting with continuous action space and concave utility function is different from the $K$-armed bandits setting of Agarwal et al. (2021) and linear utility function case of Di et al. (2024). Therefore, neither their lower bounds nor their proof techniques are directly applicable to our setting with corruption to strictly concave utilities; see more discussions and comparisons below.

Our second main result is an efficient algorithm with regret upper bound $\tilde{\mathcal{O}}(d \max\{\sqrt{T}, T^{\rho}\})$ that matches the above lower bound, up to logarithmic terms, in the same setting (i.e., decaying corruption level $t^{\rho-1}$ with known $\rho$). Unlike previous corruption-robust dueling bandit algorithms based on robust $K$-armed bandits (Agarwal et al., 2021) or robustified LinUCB (Di et al., 2024), our algorithm is a gradient-based algorithm that is built upon the recent Noisy Comparison-based Stochastic Mirror Descent (NC-SMD) of Kumagai (2017). This is natural for our setup with continuous actions and strictly concave utility. While our algorithm itself can be viewed a natural generalization of Kumagai (2017), its analysis is highly non-trivial and much more complex, due to the necessity of accounting for corruption.

Our main technical contribution is a new framework for analyzing the regret of gradient-based algorithms under arbitrary corruption. This framework introduces a novel regret decomposition lemma for dueling bandits, which upper bounds total regret by combining decision regret and feedback error due to corruption, based on quantifying bias in gradient estimation. It yields a tight analysis of our new algorithm and implies regret bounds for popular existing algorithms in (even fully) adversarial corruption settings, such as the dueling bandit gradient descent (DBGD, Yue & Joachims (2009)).

These upper bounds are looser than our lower bound for the LIHF setting. It is an intriguing open question to understand whether these upper bounds under arbitrary corruption are tight, or the lower bound under arbitrary corruption may be stronger than our lower bound under LIHF. Finally, we conduct simulations which validate our theory.

**Comparisons with Related Works.** Since multiple recent works (Agarwal et al., 2021; Saha & Gaillard, 2022; Di et al., 2024) study dueling bandits under arbitrarily adversarial corruption , it is worthwhile to discuss the key difference between these works and ours. The first key difference is the setting, which leads to fundamentally different algorithms hence analysis. Specifically, our setting with continuous action space and a completely unknown concave utility function is motivated by addressing problems from modern recommender systems where the action (i.e., contents) space is extremely large, not enumerable, thus often embedded as high-dimensional continuous vectors (Chen et al., 2019; 2021). This setting and our techniques are crucially different from earlier works. Agarwal et al. (2021); Saha & Gaillard (2022) consider the $K$-arm dueling bandit setting and utilizes

the reduction of Ailon et al. (2014) from dueling bandits to multi-$K$-armed bandits (MAB); (Di et al., 2024) considers linear utility setting where the unknowns is a parameter vector $\theta$, and utilizes techniques from linear contextual bandits based on the optimism principle. The methods all rely on parameter estimation which are not applicable to our setting with arbitrary concave utilities. Moreover, they often need to enumerate all actions to identify the best one under optimism, which are not computationally efficient for our setting with continuous action space. This is why our algorithm is gradient-based and falls within the family of stochastic mirror descent. To the best of our knowledge, this is the first time a corruption-robust algorithm is developed for dueling bandits with continuous actions and strictly concave utilities, which is also why it is necessary for us to develop a new analysis framework. Additionally, we also note that dueling bandits with strictly concave utility and linear utility are generally not comparable, both in terms of their problem difficulties and methodologies. A strictly more general situation is the concave utility case (no need to be strongly concave), however the best upper bound so far for general concave utility is $\mathcal{O}(T^{3/4})$ (Flaxman et al., 2005; Yue & Joachims, 2009), which are considerably worse than the $\mathcal{O}(T^{1/2})$ for linear and for strongly concave utility.

The second main difference between our work and previous is the more restricted class of adversarial environments. Our model is motivated by imperfect human feedback whereas previous studies are motivated by arbitrarily adversarial adversaries. Finally, a more subtle difference is that the corruption in our model is on utility with motivations from learning from imperfect human feedback, whereas corruption in all previous settings directly flip the comparison outcome. These two differences lead to a very different proof techniques of our lower bound. Our proof is more involved because the adversary has limited capability to alter the comparison outcomes. Specifically, the corruption are decaying; moreover, a constant amount of utility corruption only slightly shifts the outcome probability whereas in previous settings each corruption completely changes the outcome.

## 2 THE PROBLEM OF LEARNING FROM IMPERFECT HUMAN FEEDBACK

**Notation.** For a positive integer $T$, we use $[T]$ to denote $\{1, 2, \ldots, T\}$. We use standard asymptotic notations including $\mathcal{O}(\cdot)$, $\Omega(\cdot)$, $\Theta(\cdot)$. We use $\tilde{\mathcal{O}}(\cdot)$, $\tilde{\Omega}(\cdot)$, $\tilde{\Theta}(\cdot)$ to hide logarithmic factors. We use $\|\cdot\|_2$ to define $L_2$ norm, $\|\cdot\|_\infty$ to define infinity norm, and $\lambda_{\max}(\cdot)$ to denote maximum eigenvalue.

Motivated by alignment of machine learning models with human preferences (Bai et al., 2022), we study learning from comparative human feedback within the seminal dueling bandit framework (Yue & Joachims, 2009) but account for "imperfections" in human feedback, as described below.

**Basics of Continuous Dueling Bandits.** We consider the dueling bandit framework with continuous *action* set $\mathcal{A} \subset \mathbb{R}^d$ (Yue & Joachims, 2009). At each round $t \in [T]$, the learner chooses two actions $a_t, a_t'$ to present to some human agent, henceforth denoted as the "user". The user receives *utility* $\mu(a_t), \mu(a_t')$ from the two actions respectively, and will pick one of these actions following a *link* function $\sigma(\cdot)$. In the absence of corruption, the user selects action $a_t$ with probability $\sigma(\mu(a_t) - \mu(a_t'))$ and chooses action $a_t'$ otherwise. Since the comparison outcome has less errors compared to the exact utility value while still carries useful information about the underlying utility function $\mu$, this form of human feedback has been widely used for learning human preferences (Ailon et al., 2014; Maystre & Grossglauser, 2017; Bengs et al., 2021; Bai et al., 2022). Following standard assumptions in this field (Yue & Joachims, 2009; Kumagai, 2017), we also assume

1. the action set $\mathcal{A}$ is a convex compact set that contains the origin, has non-empty interior, and is contained in a $d$-dimensional ball of radius $R$;

2. the utility function $\mu : \mathcal{A} \to \mathbb{R}$ is strongly concave and twice-differentiable. The following constants are useful for our algorithm analysis: $\mu$ is $L^\mu$-Lipschitz, $\alpha$-strongly concave,[1] $\kappa$-smooth, bounded, $R^\mu := \sup_{a,a' \in \mathcal{A}} \mu(a) - \mu(a')$, and $a^* := \arg\max_{a \in \mathcal{A}} \mu(a)$ is the unique optimal within $\mathcal{A}$;

3. the link function $\sigma : \mathbb{R} \to [0, 1]$ is smooth, rotation-symmetric (i.e. $\sigma(x) = 1 - \sigma(-x)$), and concave for $\mathbb{R}^+$. Since $R^\mu$ is finite, there exists positive $l_1^\sigma$, $L_1^\sigma$ and $R_2^\sigma$ such that $l_1^\sigma \leq \sigma' \leq L_1^\sigma$ on $[-R^\mu, R^\mu]$, and the second derivative is bounded above by $R_2^\sigma$ and $L_2^\sigma$-Lipschitz on $[-R^\mu, R^\mu]$. For a more detailed discussion on the generality of the modeling assumptions, please refer to Appendix A.

---

[1] $\mu$ is $\alpha$-strongly concave for some $\alpha > 0$ if $\mu(x) \geq \mu(y) + \langle \nabla\mu(x), x - y \rangle + \frac{\alpha}{2}\|y - x\|_2^2$ for any $x, y \in \mathcal{A}$.

A broad range of natural link functions satisfy our assumptions, including the standard logistic distribution function, the cumulative standard Gaussian distribution function, and the linear function. This distinguishes our work from some prior studies, working with concrete link function formats such as logistic function (Maystre & Grossglauser, 2017; Saha, 2021; Xu et al., 2024) or the linear function (Ailon et al., 2014; Chen & Frazier, 2017; Zimmert & Seldin, 2018; Lin & Lu, 2018).

**Modeling Imperfect Human Feedback as a Restricted Form of Adversarial Corruption.** To incorporate imperfect human feedback, we study a generalization of the dueling bandit framework above, by introducing a *corruption* term, $c(a, a')$, to the utility difference. Formally, let $a \succ a'$ represent the event where the user chooses $a$ over $a'$. The probability of this event in the presence of corruption $c(a, a')$ is denoted by $\hat{\mathbb{P}}(a \succ a')$, expressed as $\hat{\mathbb{P}}(a \succ a') := \sigma(\mu(a) - \mu(a') + c(a, a'))$. We denote the user's preferential feedback under corruption as $\hat{\mathcal{F}}(a, a')$, referred to as *imperfect dueling feedback*. Mathematically, $\hat{\mathcal{F}}(a, a')$ follows a binomial distribution with mean $\hat{\mathbb{P}}(a \succ a')$, as expressed by the following equation.

$$\hat{\mathcal{F}}(a, a') := \begin{cases} 1 \text{ w.p. } \hat{\mathbb{P}}(a \succ a') \\ 0 \text{ w.p. } 1 - \hat{\mathbb{P}}(a \succ a') \end{cases}$$

The "hat" notation in $\hat{\mathcal{F}}$ and $\hat{\mathbb{P}}$ is to emphasize the presence of corruption. When there is no corruption, we use $\mathbb{P}(a \succ a') := \sigma(\mu(a) - \mu(a'))$ to represent the probability of the event $a \succ a'$, and $\mathcal{F}(a, a')$ to denote the corresponding dueling feedback.

We remark that the term $c(a, a')$ above is often called "strong corruption" in the literature of adversarial corruption because it can be introduced after observing actions $a$ and $a'$ and the corruption function, $c(\cdot)$, can depend on $\mu$ and past actions (Bogunovic et al., 2020; He et al., 2022; Di et al., 2024). The only difference between our model and the above works on adversarial corruption is a natural restriction to the scale of the corrupted term $c(a, a')$. Our motivation is that human feedback, while imperfect, should not be arbitrarily adversarial and tends to improve as human agent interact with the learner. This motivates our following definition of *imperfect human feedback*.

**Definition 2.1** ($\rho$-Imperfect Human Feedback). *The human feedback is said to be $\rho$-imperfect for some $\rho \in [0, 1]$ if there exists a constant $C_\kappa$ such that corruption $c_t(a_t, a'_t)$ satisfies $|c_t(a_t, a'_t)| \leq C_\kappa t^{\rho-1}, \forall t \in [T]$.*[2]

The definition shows that $\rho$-imperfect human feedback is mathematically equivalent to a restricted form of strong adversarial corruption to user utility functions. The total corruption is bounded by $\sum_{t=1}^{T} |c_t(a_t, a'_t)| \leq C_\kappa T^\rho$, which is a key parameter influencing the intrinsic difficulty of the learning problem. We thus cast the Learning from Imperfect Human Feedback (LIHF) problem as dueling bandits under strong corruption but with decaying scales. For clarity, we use "arbitrary adversarial corruption" as corruption without decaying constraints and "$\rho$-Imperfect Human Feedback" as corruption with decaying constraints, as described in Definition 2.1. Through this modeling, we aim to: first create a more optimistic framework for LIHF compared to arbitrary adversarial corruption, and second, explore LIHF's intrinsic difficulty to achieve more efficient learning guarantees than those possible under arbitrary adversarial corruption. We direct readers to Appendix B for a more comprehensive discussion on challenges in developing robust algorithms for continuous dueling bandits with generally concave utility under arbitrary adversarial corruption.

Several points are worth noting about Definition 2.1. While we are not the first to model imperfect human feedback, we are the first to apply it to dueling bandits with strictly concave utilities. Recent works, motivated by recommendation systems, have explored learning user preferences from imperfect feedback (Immorlica et al., 2020; Yao et al., 2022; Wang et al., 2023). These studies assume that users do not know their true expected reward $\theta_i$ for each choice $i$ (i.e., arm) and instead behave based on an estimated reward $\hat{\theta}_{t,i}$ at time $t$. The utility difference $|\theta_i - \hat{\theta}_{t,i}| = c_t$ is modeled as a decreasing function of time, $t^{\rho-1}$, indicating that users improve their preference estimates over time. By viewing the human utility $\mu(a)$ as the average reward, our modeling of $\rho$-Imperfect Human Feedback shares the same spirit; it captures imperfect yet gradually improving human feedback (e.g., due to inaccuracies in utility perception or initial ignorance of true preferences). As one of our motivations, when generative models like ChatGPT learn to create personalized content, the user

---

[2] All our results naturally applies to $\rho < 0$, which is a significantly easier situation since accumulated corruption is a constant in that case. Therefore, we will not explicitly consider it in this paper.

could undergo a process of preference refinement during interactions with the model. Additionally, our model differs from recent corruption-robust dueling bandit studies (Komiyama et al., 2015; Saha & Gaillard, 2022; Di et al., 2024), which corrupt realized outcomes (e.g., flipping $a' \succ a$ to $a \succ a'$) and are motivated by adversaries like malicious users or fraudulent clicks (Deshpande & Montanari, 2013; Lykouris et al., 2018). In contrast, our model focuses on utility corruption caused by imperfect perception of true utility, not outcome manipulation.

**Learning goal: regret minimization by Learning from $\rho$-Imperfect Human Feedback ($\rho$-LIHF).** The goal of the learner is to optimize her sequential actions to minimize the following dueling regret in the $\rho$-LIHF problem:

$$\text{Reg}_T := \mathbb{E}\left(\sum_{t=1}^{T} \sigma(\mu(a^*) - \mu(a_t)) + \sigma(\mu(a^*) - \mu(a_t')) - 1\right). \tag{1}$$

This regret measure has been extensively studied in prior literature (Yue & Joachims, 2009; Komiyama et al., 2015; Kumagai, 2017; Saha & Gaillard, 2022). Other regret measures, such as the cumulative difference between optimal and obtained utility, are also considered. We discuss the equivalence of various regret measures in Lemma D.1.

## 3 THE INTRINSIC LIMIT OF LIHF

In this section, we study the intrinsic limit of learning from $\rho$-Imperfect Human Feedback. We are particularly interested in understanding whether learning from this restricted version of corrupted feedback is fundamentally "easier" than learning from arbitrarily adversarial corruption. Somewhat surprisingly, the answer seems to be *no*, as illustrated by the following lower bound result.

**Theorem 3.1.** *There exists a $\rho$-Imperfect Human Feedback (see Definition 2.1), strongly concave utility function $\mu$, and link function $\sigma$ such that any learner has to suffer $Reg_T \geq \Omega\left(d \max\{\sqrt{T}, T^\rho\}\right)$, even with the knowledge of $\rho$.*

Notably, similar lower bound results have appeared in recent studies on adversarial corruption in bandits (Bogunovic et al., 2022), including dueling bandits (Agarwal et al., 2021; Di et al., 2024), showing the same lower bound $\Omega(d \max\{\sqrt{T}, T^\rho\})$ (though they often use $C := \Theta(T^\rho)$ to denote total corruption budget). However, a few key distinctions are worth noting. First, Theorem 3.1 presents a stronger result, showing that even when corruption decays over time, this structure (which defines our LIHF problem) does not make learning easier. Second, our corruption targets utility values, which is quite different from the corruption of comparison outcomes studied in (Agarwal et al., 2021; Di et al., 2024). In their case, corruption can fully flip comparison outcomes, which is critical for promoting sub-optimal arms in their lower bound proofs. However, such full control of the outcome is infeasible in our corruption of only the utility, which only slightly shifts the comparison probabilities since the scale of the corruption is also upper bounded by $\mathcal{O}(t^{\rho-1})$.

The aforementioned difference from previous works (Agarwal et al., 2021; Di et al., 2024) also render our proof of Theorem 3.1 fundamentally different from (and much more involved than) their proofs. Specifically, due to limited and diminishing corruption power, the corruption in our LIHF problem are insufficient to completely "mask" the instance as one with a different sub-optimal arm, which is the key strategy in previous lower bound proofs. Thus, our proof has to leverage information-theoretic lower bounds of statistical distributions to understand how small-scale corruption at each round could influence the overall function estimation errors. Core of our proof is the follow lower bound result for a different, and intuitively easier, problem: the standard bandit setup with direct reward feedback shown in Lemma 3.1 below. We then convert this lower bound for direct reward feedback to dueling feedback through a linear link function, i.e. $\sigma(x) = \frac{1+x}{2}$.

**Lemma 3.1** (Lower Bound under Direct Reward Feedback). *Consider bandit learning with direct reward feedback, where reward $r(a) := \mu(a) + \epsilon$, $\epsilon$ follows standard normal distribution. The action space $\mathcal{A}$ is contained in a $d$-dimensional unit ball, and $\mu$ is $\mu_\theta(a) := \theta^\top a - \frac{1}{2}\|a\|_2^2$, with $\theta \in \mathbb{R}^d, \|\theta\|_2 \leq 1$. Under $\rho$-Imperfect Human Feedback (Definition 2.1), for any $T$, $d \leq \frac{1}{C_\kappa}T^{1-\rho}$, and for any learner, there exists a $\theta$ such that under direct reward feedback, even with the knowledge of $\rho$[3], has to suffer regret $Reg_T := \mathbb{E}\{\sum_{t=1}^{T} \mu_\theta(a^*) - \mu_\theta(a_t)\} \geq \frac{d}{4}C_\kappa T^\rho$.*

---

[3]We only consider $\rho \geq \frac{1}{2}$. If $\rho < \frac{1}{2}$, the lower bound degenerates to $\Omega(d\sqrt{T})$ (Kumagai, 2017).

*Proof Sketch of Lemma 3.1.* We aim to show that there exists a corruption strategy such that the regret incurred by any learner with direct reward feedback is no less than $\Omega(dT^\rho)$. The formulation of such a corruption strategy hinges on the observation that the regret incurred by the learner is bounded below by the sum of the Kullback-Leibler (KL) divergence between the distribution of the corrupted reward feedback $\hat{v}_t$ conditioned on different values of $\theta$. Specifically, let $\theta$ be uniformly drawn from $\{-\beta, \beta\}^d$, for some constant $\beta > 0$. Given an action $a_t$ and index $i$, conditioned on the value of the $i$-th coordinate of $\theta$, $\theta_i$, if $\theta_i > 0$, then $\hat{v}_t$ is

$$\hat{v}_t = \mu_\theta(a_t) + c_t(a_t|\theta_i > 0) + \xi_{a_t} = \left( -\frac{1}{2}\|a_t\|^2 + \sum_{j \neq i} \theta_j a_{t,j} \right) + \beta a_{t,i} + c_t(a_t|\theta_i > 0) + \xi.$$

Conditioned on $\theta_i < 0$, the corrupted reward feedback $\hat{v}_t'$ is

$$\hat{v}_t' = \mu_\theta(a_t) + c_t(a_t|\theta_i < 0) + \xi_{a_t} = \left( -\frac{1}{2}\|a_t\|^2 + \sum_{j \neq i} \theta_j a_{t,j} \right) - \beta a_{t,i} + c_t(a_t|\theta_i < 0) + \xi.$$

The noise $\xi$ follows standard normal distribution. Consider the following corruption strategy which sets $c_t(a_t|\theta_i > 0) := -\beta a_{t,i}$ and $c_t(a_t|\theta_i < 0) := \beta a_{t,i}$ to minimizes the KL divergence between $\hat{v}_t$ and $\hat{v}_t'$. Together with 1-strongly concavity of $\mu_\theta$, we can establish $\mathbb{E}\left\{\sum_{t=1}^T \mu_\theta(a^*) - \mu_\theta(a_t)\right\} \geq \frac{1}{2}\max\left\{\sum_{t=1}^T (\frac{C_\kappa T^{\rho-1}}{\beta} - \beta)^2, \frac{dT\beta^2}{2}\right\}$. Since $d \leq \frac{1}{C_\kappa}T^{1-\rho}$, we can set $\beta := \sqrt{C_\kappa}T^{\frac{\rho-1}{2}}$ to satisfy $\|\theta\|_2 \leq 1$ and to optimize the aforementioned lower bound. Then we obtain $\mathbb{E}\left\{\sum_{t=1}^T \mu_\theta(a^*) - \mu_\theta(a_t)\right\} \geq dC_\kappa T^\rho/4$. Notice that the corruption level each round is bounded by $\beta\|a_t\|_\infty$. In the scenario when the radius of $\mathcal{A}$ is upper bounded by $\beta$, the corruption budget each round is bounded by $C_\kappa t^{\rho-1}$. It implies that the corruption is $\rho$-Imperfect Human Feedback, completing the proof. □

Theorem 3.1 can be proved by converting the lower bound in Lemma 3.1 for direct reward feedback to dueling feedback (see Lemma C.1). Before concluding this section, we note that the strongly concave utility function in the lower bound of Theorem 3.1 is not necessary. In particular, Proposition 1 below shows that similar lower bound applies to linear utilities as well. This result happens to resolve an open problem posed by Yao et al. (2022) which studies a similar learning from imperfect user problem with linear utility and dueling feedback (they termed it the "learning from a learning user" problem), develops a no-regret learning algorithm for that setting, but left the lower bound as an open problem. Proposition 1 implies that their upper bound's dependence on $T$ is tight. We direct readers to Appendix C.1 and C.2 for proof details of Lemma 3.1 and Proposition 1.

**Proposition 1.** *There exists an LIHF instance with $\rho$-Imperfect Human Feedback (Definition 2.1), linear user utility $\mu$, and link function $\sigma$, such that any learner has to suffer $Reg_T \geq \Omega(d\max\{\sqrt{T}, T^\rho\})$, even with the knowledge of $\rho$.*

## 4 AN EFFICIENT AND TIGHT $\rho$-LIHF ALGORITHM

In this section, we present a learning algorithm with a regret upper bound that matches the $\Omega(d\max\{\sqrt{T}, T^\rho\})$ lower bound, up to a logarithmic term, for the $\rho$-LIHF setting discussed in Section 3. Our algorithm, **Ro**bustified **S**tochastic **M**irror Descent for **I**mperfect **D**ueling (RoS-MID), outlined in Algorithm 1, generalizes the Noisy Comparison-Based Stochastic Mirror Descent (NC-SMD) from Kumagai (2017). RoSMID employs a corruption-aware learning rate, $\eta_\rho := \sqrt{\log T}/(dT^{\max\{0.5,\rho\}})$, to slow down the learning when the imperfection level $\rho$ is large, with NC-SMD being the special case with $\rho = 0$. Although RoSMID may initially seem like a pretty natural extension of NC-SMD, the choice of optimal $\eta_\rho$ is in fact through careful derivation from a novel framework for analyzing continuous bandits under utility corruption. We conclude by demonstrating the applicability of this new framework to analyzing other gradient-based algorithms.

**Theorem 4.1.** *RoSMID satisfies $Reg_T \leq \tilde{\mathcal{O}}(d\max\{\sqrt{T}, T^\rho\})$ for any $\rho$-LIHF problem.*

The main technical contribution underlying this theorem is a novel framework for analyzing continuous dueling bandits with *arbitrary strongly concave*, yet possibly imperfect or corrupted utilities, which we believe has independent value. This framework not only enables a tight regret upper bound analysis, as shown in Theorem 4.1, but also allows us to easily derive regret guarantees for other dueling bandit algorithms under adversarial corruption, as demonstrated in Subsection 4.1. While recent works have explored corruption-robust regret analysis for dueling bandits with $K$ arms

---

**Algorithm 1: R**obustified **S**tochastic **M**irror Descent for **I**mperfect **D**ueling (**RoSMID**)

---

**Input:** Learning rate $\eta_\rho := \sqrt{\log T} / (dT^{\max\{0.5,\rho\}})$, $\nu$-self-concordant function $\mathcal{R}$ [4], time

horizon $T$, tuning parameters $\lambda \leq \frac{l_1^\sigma \alpha}{2}$, $\phi \geq ((L_1^\sigma)^3 L_2^\sigma / \lambda)^2$

1 Initialize $a_1 = \arg\min_{a \in A} \mathcal{R}(a)$
2 **for** $t \in [T]$ **do**
3      Update concordant function $\mathcal{R}_t(a) = \mathcal{R}(a) + \frac{\lambda \eta_\rho}{2} \sum_{i=1}^t \|a - a_i\|^2 + \phi \|a\|^2$
4      Sample direction $u_t$ uniformly at random from a *unit sphere* $\mathbb{S}^d$
5      Obtain corrupted feedback $\hat{\mathcal{F}}(a_t', a_t) \in \{0, 1\}$, for $a_t$ and $a_t' = a_t + \nabla^2 \mathcal{R}_t(a_t)^{-1/2} u_t$
6      Compute the corrupted gradient: $\hat{g}_t = \hat{\mathcal{F}}(a_t', a_t) d \cdot \nabla^2 \mathcal{R}_t(a_t)^{1/2} \cdot u_t$
7      Set $a_{t+1} = \nabla \mathcal{R}_t^{-1}(\nabla \mathcal{R}_t(a_t) - \eta_\rho \hat{g}_t)$

**Output:** $a_{T+1}$

---

(Saha & Gaillard, 2022) or linear utilities (Di et al., 2024), neither their algorithms nor analysis can be applied to our setting with arbitrary concave utilities in continuous action space.

The full proof of Theorem 4.1 is quite involved and is deferred to Appendix D. Here, we outline the main ideas, divided into four key steps. Step 1-3 are applicable for analyzing continuous dueling bandits under any form of corruption, whereas last Step 4 is the only step that hinges on the decaying corruption level assumptions of $\rho$-LIHF and is also the most involved and novel step.

*Proof Sketch.* The proof of Theorem 4.1 is divided into four major steps.

**Step 1: quantifying bias of gradient estimation in $\rho$-LIHF**

Our proof starts from quantifying the bias of the gradient $\hat{g}_t$ caused by $\rho$-Imperfect Human Feedback, as shown in the following Lemma 4.1.

**Lemma 4.1** (Corrupted Gradients Estimation). *The gradient $\hat{g}_t$ in Line 6 of Algorithm 1 satisfies*

$$\mathbb{E}(\hat{g}_t | a_t) = \nabla|_{a=a_t} \bar{P}_t(a) - \mathbb{E}(b_t | a_t).$$

*where $\bar{P}_t(a) := \mathbb{E}_{x \in \mathbb{B}} \left[ \mathbb{P}(a_t \succ a + \nabla^2 R_t(a_t)^{-\frac{1}{2}} x) \right]$ (with $\mathbb{B}$ as the unit ball) is the smoothed probability that $a_t$ is preferred over any action $a$, and $b_t = d \left( \mathbb{P}(a_t \succ a_t') - \hat{\mathbb{P}}(a_t \succ a_t') \right) \nabla^2 \mathcal{R}_t(a_t)^{\frac{1}{2}} u_t$.*

**Step 2: decomposing regret for dueling bandits into regret of decision and feedback error**

Core to our proof is a "regret decomposion" lemma below that upper bounds the regret by two parts: regret of sub-optimal decision making under uncertainty, referred as regret of decision (first term in equation 2), and regret due to corruption, referred as feedback error (second term in equation 2).

**Lemma 4.2** (Regret Decomposition for Dueling Bandits). *The $Reg_T$ of Algorithm 1 satisfies:*

$$Reg_T \leq \underbrace{\frac{C_0}{\eta_\rho} \log(T) + 8d^2 \eta_\rho T + 2L^\mu L_1^\sigma R}_{\textit{Regret of Decision}} + \underbrace{2\mathbb{E} \left\{ \sum_{t=1}^T b_t^\top (a_t - a_T^*) \right\}}_{\textit{Feedback Error}}, \tag{2}$$

*where $C_0 := (2\nu + 4L_1^\sigma \kappa + 4R_2^\sigma (L^\mu)^2 + L^\mu L_1^\sigma \kappa)/\lambda$, and $a_T^* := \frac{1}{T} a_1 + (1 - \frac{1}{T}) a^*$.*

Similar regret decomposition has been shown to be useful for reinforcement learning which has direct reward feedback (Foster et al., 2023). However, to the best of our knowledge, Lemma 4.2 is the first to exhibit such a decomposition for dueling feedback, which contains much sparser information than direct reward feedback as in classic RL or online learning. Hence the lemma may be of independent interest for future research on dueling bandits under corruption or erroneous feedback.

**Step 3: upper bounding the regret from feedback error**

Bounding the regret of decision term in equation 2 is standard. Thus, this step develops a novel upper bound for bounding the regret from feedback error, as shown below.

**Lemma 4.3** (Feedback Error). *The feedback error term in equation 2 can be bounded as follows:*

$$\mathbb{E}\left\{\sum_{t=1}^{T} b_t^\top (a_t - a_T^*)\right\} \le C_1 d\sqrt{\eta_\rho}\mathbb{E}\left\{\sum_{t=1}^{T} \sqrt{t}|c_t(a_t, a_t')|\sqrt{\mu(a^*) - \mu(a_t)}\right\} + C_2 dT^\rho + 2d\sqrt{\lambda}\sqrt{\eta_\rho T},$$

*where $C_1 := \sqrt{\frac{2\lambda(L^\sigma)^2}{\alpha}}$, and $C_2 := 2L^\sigma R\sqrt{\sup_{a\in\mathcal{A}} \lambda_{\max}(\nabla^2 \mathcal{R}(a)) + 2\phi}$.*

To our knowledge, such a bound is new for dueling bandits. The less common term in the bound is $\sqrt{\mu(a^*) - \mu(a_t)}$. It comes from the strong concavity of the utility function $\mu$, which implies $\langle b_t, a_t - a^*\rangle \le \|b_t\|_2\|a_t - a^*\|_2 \le \|b_t\|_2\sqrt{\frac{2}{\alpha}(\mu(a^*) - \mu(a_t))}$.

**Step 4: The *iterative regret refinement* analysis and resultant optimal learning rate**

Our last and also the most intriguing step is to pin down the regret upper bound. Given the bounds from previous steps, the only remaining term to upper bound is $\mathbb{E}\left\{\sum_{t=1}^{T} \sqrt{t}|c_t(a_t, a_t')|\sqrt{\mu(a^*) - \mu(a_t)}\right\}$ in Lemma 4.3. We can upper bound $\sqrt{t}|c_t(a_t, a_t')|$ by $C_\kappa t^{\rho-\frac{1}{2}}$ under $\rho$-Imperfect Human Feedback assumption (this is the point where we start to need the decaying structure in Definition 2.1). This is a tricky question, since this term involving optimal action $a^*$ closely connects to the regret $\text{Reg}_T$ itself (see equation 1). This is precisely the motivation for our analysis approach of "iterative regret refinement". Concretely, it turns out that a good upper bound for the term $A := \mathbb{E}\left(\sum_{t=1}^{T} \mu(a^*) - \mu(a_t)\right)$ can be leveraged to derive a good upper bound for term $B := \mathbb{E}\left\{\sum_{t=1}^{T} t^{\rho-\frac{1}{2}}\sqrt{\mu(a^*) - \mu(a_t)}\right\}$, which can then be leveraged to derive a good upper bound for the regret $\text{Reg}_T$, which can be used to refine the original upper bound of term $A$, at which point we can re-apply the above refinement process.

Two key factors are essential for the iterative regret refinement mentioned above to result in a tight bound. The first step is to derive a good upper bound for term $B$ using term $A$, which is shown in Lemma 4.4. The second step is to prove that this iterative refinement consistently improves the regret bound, eventually reaching a limit. This is demonstrated by Lemma 4.5. The choice of learning rate $\eta_\rho = \frac{\sqrt{\log T}}{dT^{\max\{1/2, \rho\}}}$ is crucial in the second step; in a sense, it is the only choice that converges to the tight regret upper bound $\tilde{\mathcal{O}}\left(d\max\{\sqrt{T}, T^\rho\}\right)$ in the limit, rather than exploding the bound.

**Lemma 4.4.** *If $A = \mathbb{E}\left(\sum_{t=1}^{T}[\mu(a^*) - \mu(a_t)]\right) \le \mathcal{O}(T^{\frac{1}{2}+\psi})$ for some $\psi \in [0, \frac{1}{2})$, then we must have $B = \mathbb{E}\left(\sum_{t=1}^{T} t^{\rho-\frac{1}{2}}\sqrt{\mu(a^*) - \mu(a_t)}\right) \le \mathcal{O}(T^{-\frac{1}{4}+\frac{\psi}{2}+\rho})$.*

To prove Lemma 4.4, we first bound term $B$ by $\sum_{t=1}^{T} t^{\rho-1/2}\sqrt{\mathbb{E}[\mu(a^*) - \mu(a_t)]}$ using the concavity of the square root function. The key novelty of our proof is to use the Abel's equation (Williams, 1963) to re-arrange the term $\sum_{t=1}^{T} t^{\rho-1/2}\sqrt{\mathbb{E}[\mu(a^*) - \mu(a_t)]}$ and establish its connection with another term $C = \sum_{t=1}^{T} \sqrt{\mathbb{E}[\mu(a^*) - \mu(a_t)]}$. We finally connect term $C$ with the term $A$ in the lemma statement, by proving $\sum_{t=1}^{T} \sqrt{\mathbb{E}[\mu(a^*) - \mu(a_t)]} \le \mathcal{O}(T^{\frac{3}{4}+\frac{\psi}{2}})$ if $\mathbb{E}\left(\sum_{t=1}^{T} \mu(a^*) - \mu(a_t)\right) \le \mathcal{O}(T^{\frac{1}{2}+\psi})$ via concavity of square root. Together with the decay scale $t^{\rho-1}$, which allows $t^{\rho-1} - (t+1)^{\rho-1} \le t^{\rho-2}$, we obtain the tight bound $\mathcal{O}(T^{-\frac{1}{4}+\frac{\psi}{2}+\rho})$.[5]

Armed with Lemma 4.4, we can use it to prove the induction claim (Lemma 4.5). The challenge in this proof lies in identifying the constant $K$, which must hold across all iterations, $k$, of the analysis to ensure the bound remains stable and does not explode.

**Lemma 4.5** (Induction Claim). *Consider $T > \sqrt{2L^\mu L_1^\sigma R}$ and $\eta_\rho = \frac{\sqrt{\log T}}{dT^{\max\{1/2, \rho\}}}$. If $\text{Reg}_T \le 144dK\sqrt{\log T}T^{\rho+\frac{\frac{3}{2}-\rho}{2^k}}$ for some integer $k$, then we must also have*

---

[5]If corruption $c_t$ was arbitrary, then $t^{\rho-1/2}$ is $\sqrt{t}|c_t|$, and the bound could deteriorate to $\mathcal{O}(T^{\frac{2\rho}{3}})$ in the first $T^\rho$ rounds.

$Reg_T \leq 144dK\sqrt{\log T}T^{\rho + \frac{\frac{3}{2} - \rho}{2^{k+1}}}$. $K$ is a instance-dependent constant, where $K :=$ $\max\left\{C_0, \frac{C_2 C_\kappa}{\sqrt{\log T}}, (L^\sigma C_\kappa)^2, 2\sqrt{\lambda}RC_\kappa L^\sigma\right\}$, $C_0$ and $C_2$ defined in Lemma 4.2 and 4.3 respectively.
□

## 4.1 ADDITIONAL APPLICATIONS OF OUR REGRET ANALYSIS FRAMEWORK

To showcase useful applications of the new regret analysis framework above for proving Theorem 4.1, in this subsection we employ the framework to study continuous dueling bandits under *arbitrary* and *agnostic* corruption — i.e., remove decaying constraints on $c_t$ and $\rho$ is unknown to the learner. We analyze natural variants of RoSIMD, which apply to strongly concave utilities, and the well-known algorithm Dueling Bandit Gradient Descent (DBGD) (Yue & Joachims, 2009), which apply to general concave utilities. The proofs of these results are direct applications of the analysis from Steps 1-3 outlined above (without Step 4, since it is the only one requiring knowledge of $\rho$ and decaying corruption). Additionally, we present these results to offer a set of principled benchmark algorithms for the experiments in Section 5. One thing to highlight is that our robustified algorithms retain the same computational complexity as the originals, ensuring added robustness without compromising efficiency. For further details, see Appendix G.1.

**Proposition 2** (Efficiency-Robustness Tradeoff in RoSMID). *If the total corruption level satisfies* $\sum_{t=1}^T |c_t(a_t, a'_t)| \leq \mathcal{O}(T^\rho)$, *then for any* $\alpha \in [\frac{1}{2}, 1)$, *if the utility function is strongly concave, for a sufficiently large round* $T$, *by choosing learning rate* $\eta = \frac{\sqrt{\log T}}{2d}T^{-\alpha}$

1. *(Robustness) RoSMID incurs* $Reg_T \leq \tilde{\mathcal{O}}(dT^\alpha + \sqrt{d}T^{\frac{1}{2}(1-\alpha)+\rho} + dT^\rho)$.

2. *(Efficiency) There exists a strongly concave utility function* $\mu$ *and a link function* $\sigma$ *such that* $Reg_T$ *suffered by RoSMID is at least* $\Omega(T^\alpha)$ *in scenario without corruption.*

**Proposition 3** (Efficiency-Robustness Tradeoff in DBGD). *If the total corruption level satisfies* $\sum_{t=1}^T |c_t(a_t, a'_t)| \leq \mathcal{O}(T^\rho)$, *then for any* $\alpha \in (0, \frac{1}{4}]$, *if utility function* $\mu$ *is generally concave, for a sufficiently large round* $T$, *choosing* $\gamma = \frac{R}{\sqrt{T}}$ *and* $\delta = \frac{\sqrt{2Rd}}{\sqrt{13L^\sigma L^\mu T^\alpha}}$ *for DBGD*

1. *(Robustness)* $Reg_T$ *incurred by DBGD satisfies* $Reg_T \leq \mathcal{O}(\sqrt{d}T^{1-\alpha} + \sqrt{d}T^{\alpha+\rho})$.

2. *(Efficiency) There exists a linear utility function* $\mu$ *and a link function* $\sigma$ *such that* $Reg_T$ *suffered by DBGD is at least* $\Omega(T^{1-\alpha})$ *in scenario without corruption.*

Both propositions illustrate the tradeoff between *learning efficiency* and *robustness* to adversarial corruption, achieved through their tunable learning rate $\eta_\rho, \gamma, \delta$ (learning rate in DBGD is $\frac{\gamma\delta}{d}$; we direct reader to Algorithm 3 for detail). Specifically, greater tolerance for agnostic corruption (i.e., larger $\alpha$ in Proposition 2) results in worse regret in non-corrupt settings. A similar robustness-accuracy trade-off has been studied in classification problems with adversarial examples, both empirically (Tsipras et al., 2018) and theoretically (Zhang et al., 2019). However, to our knowledge, this is the first formal analysis of such a trade-off in online learning. While expanding the confidence region for UCB-type algorithms to handle agnostic corruption in linear contextual dueling bandits increases tolerance for corruption but worsens regret—sharing a similar idea of a trade-off (Di et al., 2023)—it's unclear if this approach necessarily reduces efficiency. In contrast, our work demonstrates that adjusting learning rates in gradient-based algorithms to enhance robustness against agnostic corruption inevitably decreases efficiency when no corruption is present. In particular, the efficiency statements in both propositions can be interpreted as algorithm-dependent lower bounds, concretely demonstrating that for certain problem instances, increasing robustness necessarily decreases the learning efficiency of the algorithm. In fact, the "efficiency-robustness tradeoff" is inherent under unknown adversarial attacks. Lemma E.1 in the appendix shows that algorithms that learn faster tend to be more vulnerable to unknown corruption. However, the nature of this tradeoff varies across different algorithms. It is a very intriguing and fundamental open question to study what the optimal tradeoff is under unknown adversarial corruption and how to achieve it.

## 5 EXPERIMENTS

In this section, we validate our theoretical analysis of RoSMID and DBGD under $\rho$-Imperfect Human Feedback through simulations. We also compare their performance against two baseline algo-

rithms: Doubler and the heuristic algorithm, Sparring, proposed by Ailon et al. (2014). For Sparring, we use bandit gradient descent (Flaxman et al., 2005) as the black-box bandit algorithm. We direct readers to Appendix G for implementation details.

**Experiment Setup.** We consider a standard experiment setup, which adopts a strongly concave utility $\mu_\theta(a) := \theta^\top a - \frac{1}{2}\|a\|_2^2$, and a logistic link function $\sigma(x) = \frac{1}{1+\exp(-x)}$. We choose $d = 5$, and $T = 10^5$. Our action space $\mathcal{A}$ is a $d$-dimensional ball with radius $R = 10$. The preference parameter $\theta$ is randomly sampled from the surface of $\mathcal{A}$. In our problem setting, the optimal action $a^* = \theta$ and $\mu_\theta(a^*) = 50$. We simulate $\rho$-Imperfect Human Feedback for $\rho \in [0.5, 1]$.

**Results and Discussion.** Figure 1 presents a log-log plot of regret versus iteration, $t$, which the order of regret in $T$ is represented by the slope of the line. Each line represents the average over five trials with different random seeds, and each line corresponds to a distinct value of $\rho$. The shaded region indicates $\pm$ one standard deviation. In the legend, "o" denotes the estimated order of regret. We estimate it using least squares on the last 1% of the data. Figure 1 supports Theorem 4.1, as for $\rho \in [0.5, 1)$, the estimated slope is less than $\rho$, suggesting that the choice of $\eta_\rho = \frac{\sqrt{\log T}}{dT^{\max\{1/2,\rho\}}}$ for RoSMID results in a regret bound of at most $\tilde{\mathcal{O}}(T^\rho)$.

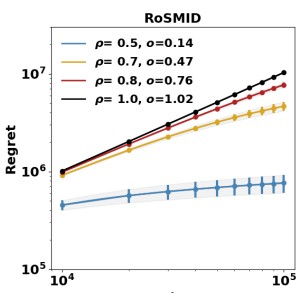

Figure 1: Robustness of RoS-MID for $\rho$-LIHF

Figure 2 compares the performance of our proposed algorithms, variants of DBGD and RoSMID, with two baseline algorithms, Sparring and Doubler, in the setting when $c_t$ is arbitrary and agnostic. We set the parameter $\alpha = 1/4$ for DBGD and $\alpha = 1/2$ for RoSMID. When $\rho$ is unknown to the algorithms, these parameter choices enable DBGD and RoSMID to tolerate $\mathcal{O}(T^{3/4})$ levels of unknown corruption, since the slope estimates are less than 1, implying sublinear regret, aligning with our theoretical predictions. Experimentally, RoSMID demonstrates the best performance under strongly concave utility functions. Sparring performs comparably to RoSMID, despite being a heuristic approach without theoretical guarantees. In contrast, Doubler exhibits the worst performance, likely because its theoretical guarantees are only applicable to linear link functions, which differs from our experimental setup. We direct readers to Appendix G for experiment results with other choice of $\alpha$ for efficiency-robustness tradeoff.

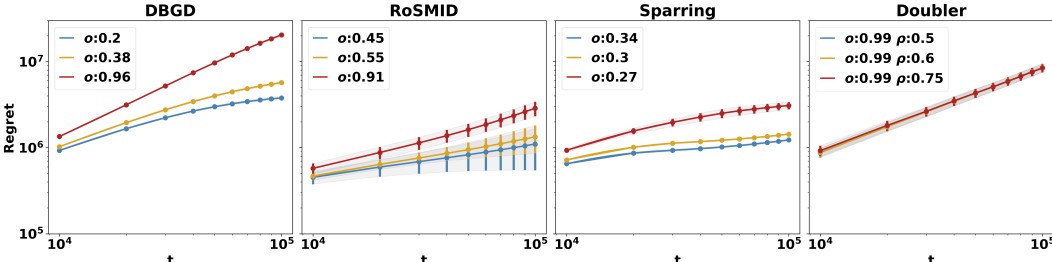

Figure 2: For each algorithm, we tested its performance under $\rho$-Imperfect Human feedback with $\rho = 0.5, 0.6, 0.75$. For each $\rho$, we presented a line plot of the average regret over five simulations, accompanied by $\pm$ one standard deviation shown by the shaded region. In the legend, $o$ denotes the estimated line slope, calculated using least squares on the last 1% of the data.

## 6 CONCLUSION

In conclusion, this paper enhances the understanding of continuous dueling bandits by addressing $\rho$-Imperfect Human Feedback with concave utilities. We established fundamental regret lower bounds, introduced the RoSMID algorithm with nearly optimal performance, and developed a novel regret analysis framework that highlights the inherent efficiency-robustness trade-off in gradient-based algorithms. Our experimental results corroborate the theoretical predictions. A promising future direction is the development of an algorithm that achieves $\tilde{O}(d\sqrt{T} + dT^\rho)$ regret for continuous dueling bandits with strongly concave utilities under arbitrary adversarial corruption. Additionally, incorporating contextual information to better model real-world scenarios could further enhance the applicability and robustness of these algorithms. We left these directions for future research.

**Acknowledgment.** This work is supported by the AI2050 program at Schmidt Sciences (Grant G-24-66104), Army Research Office Award W911NF23-1-0030, and NSF Award CCF-2303372. Fan Yao conducted this research while he is visiting the University of Chicago.

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

# Appendix to *Learning from Imperfect Human Feedback: A Tale from Corruption-Robust Dueling*

## A  DISCUSSION ON THE GENERALITY OF MODELING ASSUMPTIONS

The assumptions of a continuous action space are standard in the online convex optimization literature (Flaxman et al., 2004; Yue & Joachims, 2009; Shamir, 2013; Kumagai, 2017). Moreover, they are particularly natural for today's preference learning for recommendation systems since languages, texts, videos are mostly processed as embedding vectors which are continuous by nature. We assumed strong concave utility functions. This is standard due to at least two reasons. First, there is already a rich literature studying online optimization of strongly convex or concave functions, and our work subscribes to that rich literature (Shamir, 2013; Kumagai, 2017; Wan et al., 2022; Saha et al., 2022). Second, strongly concave utilities functions capture essential properties like diminishing marginal returns and risk-averse behavior. These characteristics align closely with real-world scenarios in economics, behavioral studies, and decision-making processes. Strongly concave functions are frequently utilized in economic literature for modeling utility functions such as the "indirect utility function" (Montrucchio, 1987; Sorger, 1995; Venditti, 1997), which represents the maximum utility a consumer can achieve based on a specific income level and set of prices. Our assumption on the link function aligns with those in Kumagai (2017) but is slightly stronger than those in Yue & Joachims (2009). Nevertheless, it encompasses many natural link functions, including the sigmoid function, the cumulative Gaussian distribution function, and the linear function, more general than merely sigmoid link functions (Maystre & Grossglauser, 2017; Saha, 2021; Xu et al., 2024) or linear link function (Ailon et al., 2014; Chen & Frazier, 2017; Zimmert & Seldin, 2018; Lin & Lu, 2018), as assumed in some earlier works.

## B  CHALLENGES IN DEVELOPING ROBUST ALGORITHMS FOR CONTINUOUS DUELING BANDITS WITH GENERALLY CONCAVE UTILITY UNDER CORRUPTION INDUCED BY STRONG ADVERSARY

There is a line of work that studies bandits under corruption by a weak adversary (Lykouris et al., 2018; Gupta et al., 2019; Agarwal et al., 2021; Zimmert & Seldin, 2021; Saha & Gaillard, 2022; Wei et al., 2022), where the corruption is introduced before observing the agent's actions. This research develops efficient algorithms that achieve sublinear regret without requiring prior knowledge of the total corruption. Specifically, for corruption induced by a weak adversary, Wei et al. (2022) adopts the idea of model selection and introduces a robust framework called Corruption-Robustness through Balancing and Elimination (COBE). This framework allows an algorithm designed for a known corruption level to be adapted for scenarios with an unknown corruption level. Notably, in the case of weak adversaries, Zimmert & Seldin (2021); Saha & Gaillard (2022) develop "best-of-both-world' algorithms, which achieve optimal performance under both begin and corrupted settings.

However, for corruption induced by a strong adversary—where the corruption occurs after observing the agent's actions (the scenario considered in our setting)—it may not be possible to develop a best-of-both-world algorithm. This limitation arises because, for any algorithm that does not know the total corruption, there exists a problem instance where the algorithm suffers linear regret (Bogunovic et al., 2020; Kang et al., 2024). To achieve sublinear regret in such settings, it is crucial for the algorithm to have precise knowledge of the total corruption or at least an accurate estimation of its upper bound.

For cases with known corruption, nearly optimal algorithms are proposed for various settings, including stochastic linear bandits in discrete action spaces (Bogunovic et al., 2020), Gaussian process bandits (Bogunovic et al., 2022), linear contextual bandits (He et al., 2022), Lipschitz bandits (Kang et al., 2024), and linear contextual dueling bandits (Di et al., 2024). These approaches typically rely on phase elimination techniques to eliminate suboptimal arms or Upper Confidence Bound (UCB)-based algorithms to pull optimal arms with high probability. The success of these methods heavily depends on constructing precise utility estimates to guide decision-making.

However, for learning from nonlinear utility, constructing utility estimates under dueling feedback is a very challenging problem, as only preferential (directional) information is available. This limited

information motivates the exploration of a new approach: robust gradient-based algorithms. Zimmert & Seldin (2021) and Saha & Gaillard (2022) utilize robust stochastic mirror descent algorithms to update the probability distribution for sampling $K$ arms. However, it is difficult to efficiently generalize this design to continuous action spaces, as it requires transforming the probability vector into a probability density function and efficiently managing its updates and approximations. While Yue & Joachims (2009) and Kumagai (2017) exploits dueling bandit gradient descent and stochastic mirror descent and achieves sublinear regret for continuous dueling bandits, their performance under corruption remains unexplored. It is a nontrivial question and still remains an interesting open problem of developing robust and computationally efficient algorithms for continuous dueling bandits with generally concave utility functions under corruption induced by strong adversaries.

## C   PROOFS FOR THEOREM 3.1

**Theorem** (Theorem 3.1 restated). *There exists a $\rho$-Imperfect Human Feedback (see Definition 2.1), strongly concave utility function $\mu$, and link function $\sigma$ such that any learner has to suffer $Reg_T \geq \Omega\left(d \max\{\sqrt{T}, T^\rho\}\right)$, even with the knowledge of $\rho$.*

*Proof.* Before showing the regret lower bound proof for Theorem 3.1, let's first establish the following Lemma C.1, which builds the connection between regret incurred under bandit reward feedback and dueling feedback. This proof is inspired by Theorem 6 in Yao et al. (2022). To prove Lemma C.1, in addition to $\text{Reg}_T$, we introduce a new metric to measure the performance of algorithm under dueling feedback, which we coin functional regret $\text{Reg}_T^{\text{FO}}$, defined as follows.

$$\text{Reg}_T^{\text{FO}} := \mathbb{E}\left\{\sum_{t=1}^{T} \mu(a^*) - \mu(a_t) + \mu(a^*) - \mu(a_t')\right\}. \tag{3}$$

**Lemma C.1.** *Given a utility function $\mu$, if the regret lower bound for algorithm with bandit reward feedback is $\overline{Reg}$, then any learner with dueling feedback has to suffer regret $Reg_T^{FO} \geq 2\overline{Reg}$.*

*Proof.* We prove our claim by contradiction. Let $(a_{0,t}, a_{1,t})$ be the pair of recommendation at round $t$. Suppose $\text{Reg}_T^{\text{FO}} < 2\overline{\text{Reg}}$. As a result, at least one of the following inequality must hold:

$$
\begin{aligned}
\mathbb{E}\left\{\sum_{t=1}^{T} \mu(a^*) - \mu(a_{0,t})\right\} &\leq \overline{\text{Reg}}; \\
\mathbb{E}\left\{\sum_{t=1}^{T} \mu(a^*) - \mu(a_{1,t})\right\} &\leq \overline{\text{Reg}}.
\end{aligned}
\tag{4}
$$

Consider a principal who can observe the interaction between a user and the learner $\mathcal{L}$, then we can construct two algorithms $\mathcal{L}_0$ and $\mathcal{L}_1$ as follows.

---
**Algorithm 2:** Algorithm $\mathcal{L}_i$

---
**Input:** the time horizon $T$

1 **for** $t \leq T$ **do**
2 $\quad$ Call the learner $\mathcal{L}$ to generate two candidates $(a_{0,t}, a_{1,t})$.
3 $\quad$ Present $(a_{0,t}, a_{1,t})$ to user and received the feedback.
4 $\quad$ Return user feedback to the learner $\mathcal{L}$ and update $\mathcal{L}$.

**Output:** the sequential decision $\{a_{i,t}\}_{t=1}^{T}$

---

From equation 4, we know that at least one of $\{\mathcal{L}_0, \mathcal{L}_1\}$ achieves an expected regret lower than $\overline{\text{Reg}}$. However, we know all algorithm with bandit feedback with utility function $\mu$ has to occur regret at least $\overline{\text{Reg}}$, contradiction. Therefore, Lemma C.1 must hold, which completes the proof. □

Next, we prove Theorem 3.1. First, consider the case $\rho \leq 0.5$. Given that the utility function is strongly concave, by applying Lemma C.2, we have $\overline{\text{Reg}} = 0.02 \min\{T, d\sqrt{T}\}$. From Lemma C.1,

we also obtain $\text{Reg}_T^{\text{FO}} \geq 0.04 \min\{T, d\sqrt{T}\}$. Using the linear link function $\sigma(x) = \frac{1}{2} + \frac{1}{2}x$, it follows that $\text{Reg}_T \geq 0.02 \min\{T, d\sqrt{T}\} \geq \Omega(d \max\{\sqrt{T}, T^\rho\})$, completing the proof.

**Lemma C.2** (Theorem 6 in (Shamir, 2013)). *Let the number of rounds $T$ be fixed. Then for any learner, there exists a quadratic function of the form $\mu_\theta(a) := \theta^\top a - \frac{1}{2}\|a\|^2$ which is minimized at $\theta$ and $\|\theta\|_2 \leq 0.5$ such that*

$$\mathbb{E}\left(\sum_{t=1}^T \mu_\theta(a^*) - \mu_\theta(a_t)\right) \geq 0.02 \min\left\{T, d\sqrt{T}\right\}.$$

Now, let's focus on the scenario when $\rho > 0.5$. In essence, we want to extend Lemma C.2 to the scenario in presence of adversarial corruption induced by $\rho$-Imperfect Human Feedback. If we can show the regret lower bound can be generalized to $\Omega(dT^\rho)$, then applying the same technique which uses Lemma C.1 to connect regret suffered under bandit reward feedback and regret suffered under dueling feedback and choose linear link function to connect $\text{Reg}_T^{\text{FO}}$ and $\text{Reg}_T$, we will get the desired regret lower bound. The extension of Lemma C.2 is proven at section C.1.

## C.1 PROOF FOR LEMMA 3.1

**Lemma** (Lemma 3.1 restated). *Consider bandit learning with direct reward feedback, where reward $r(a) := \mu(a) + \epsilon$, $\epsilon$ follows standard normal distribution. The action space $\mathcal{A}$ is contained in a $d$-dimensional unit ball, and $\mu$ is $\mu_\theta(a) := \theta^\top a - \frac{1}{2}\|a\|_2^2$, with $\theta \in \mathbb{R}^d, \|\theta\|_2 \leq 1$. Under $\rho$-Imperfect Human Feedback (Definition 2.1), for any $T$, $d \leq \frac{1}{C_\kappa}T^{1-\rho}$, and for any learner, there exists a $\theta$ such that under direct reward feedback, even with the knowledge of $\rho$, has to suffer regret $\text{Reg}_T := \mathbb{E}\{\sum_{t=1}^T \mu_\theta(a^*) - \mu_\theta(a_t)\} \geq \frac{d}{4}C_\kappa T^\rho$.*

*Proof.* To prove Lemma 3.1, we aim to prove that there exists a corruption strategy such that the regret incurred by any learner with *imperfect dueling feedback* is not less than $\Omega(dT^\rho)$. The formulation of such a corruption strategy hinges on the observation that when the utility function $\mu_\theta$ adopts the quadratic form parameterized by $\theta$, specifically $\mu_\theta(a) = \theta^\top a - \frac{1}{2}\|a\|_2^2$, which is both smooth and strongly concave, then the regret incurred by the learner is bounded below by the sum of the Kullback-Leibler (KL) divergence between the distribution of the *corrupted reward* feedback obtained at round $t$, denoted $\hat{v}_t, t \in [T]$, conditioned on different possible values of $\theta$. (see Lemma C.4). Assume that $\theta$ is uniformly drawn from $\{-\beta, \beta\}^d$, given an action $a_t$, conditioned on $\theta_i > 0$, the corrupted reward feedback $\hat{v}_t$ is

$$\hat{v}_t = \mu_\theta(a_t) + c_t(a_t|\theta_i > 0) + \xi = \underbrace{\left(-\frac{1}{2}\|a_t\|^2 + \sum_{j\neq i}\theta_j a_{t,j}\right) + \beta a_{t,i} + c_t(a_t|\theta_i > 0)}_{\mu_1} + \xi. \quad (5)$$

Conditioned on $\theta_i < 0$, the corrupted reward feedback $\hat{v}_t$ is

$$\hat{v}_t = \mu_\theta(a_t) + c_t(a_t|\theta_i < 0) + \xi = \underbrace{\left(-\frac{1}{2}\|a_t\|^2 + \sum_{j\neq i}\theta_j a_{t,j}\right) - \beta a_{t,i} + c_t(a_t|\theta_i < 0)}_{\mu_2} + \xi. \quad (6)$$

We use $c_t(a_t|\theta_i > 0)$ and $c_t(a_t|\theta_i < 0)$ to empathize the fact that the magnitude and sign of corruption can be dependent on $\theta$. $\xi$ follows a standard Gaussian distribution. Therefore, $\hat{v}_t$ in Equation 5 follows $\text{N}(\mu_1, 1)$. $\hat{v}_t$ in Equation 6 follows $\text{N}(\mu_2, 1)$. By using Lemma C.3, we have

$$\text{D}_{\text{KL}}(\text{N}(\mu_1, 1)\|\text{N}(\mu_2, 1)) = (\mu_1 - \mu_2)^2.$$

**Lemma C.3** (KL Divergence for Normal Distribution). *Let $N(\mu, \sigma^2)$ represent a Gaussian distribution variable with mean $\mu$ and variance $\sigma^2$. Then*

$$D_{KL}(N(\mu_1, \sigma^2)\|N(\mu_2, \sigma^2)) = \frac{(\mu_1 - \mu_2)^2}{2\sigma^2}.$$

To optimize the lower bound in Lemma C.4, the adversary selects $c_t(a_t|\theta_i > 0)$ and $c_t(a_t|\theta_i < 0)$ to minimize the difference between $\mu_1$ and $\mu_2$. Consider the following corruption strategy: when $\theta_i > 0$, set $c_t(a_t|\theta_i > 0) = -\beta a_{t,i}$; when $\theta_i < 0$, set $c_t(a_t|\theta_i < 0) = \beta a_{t,i}$. This strategy ensures that $\mu_1 = \mu_2$.

Next step is to determine the value of $\beta$. Notice that the corruption budget is $C_\kappa T^\rho$. To execute the corruption strategy described above, it requires

$$\sum_{t=1}^{T} |c_t(a_t)| \le \beta \sum_{t=1}^{T} \|a_t\|_\infty \le C_\kappa T^\rho. \tag{7}$$

Therefore, by exploiting the fact that $\mu_\theta$ is 1-strongly concave, we establish another lower bound for the regret by

$$\mathbb{E}\left\{ \sum_{t=1}^{T} \mu_\theta(a^*) - \mu_\theta(a_t) \right\} \ge \frac{1}{2}\mathbb{E}\left\{ \sum_{t=1}^{T} \|a_t - \theta\|_2^2 \right\}$$

$$\ge \frac{1}{2}\mathbb{E}\left\{ \sum_{t=1}^{T} (\|a_t\|_\infty - \beta)^2 \right\}$$

$$\ge \frac{1}{2}\sum_{t=1}^{T} (C_\kappa T^{\rho-1}/\beta - \beta)^2.$$

Since $|a_{t,i} - \theta_i| \ge |\|a_{t,i}\| - \beta|$ for all $i$, we obtain the second inequality. By utilizing the corruption constraint equation 7, we derive the last inequality, which is minimized when $\|a_t\|_\infty = \frac{C_\kappa T^{\rho-1}}{\beta}$ for all $t$. Together with Lemma C.4, we have:

$$\mathbb{E}\left\{ \sum_{t=1}^{T} \mu_\theta(a^*) - \mu_\theta(a_t) \right\} \ge \frac{1}{2}\max\left\{ \sum_{t=1}^{T} (C_\kappa T^{\rho-1}/\beta - \beta)^2, \frac{dT\beta^2}{2} \right\}. \tag{8}$$

We select $\beta$ to optimize the lower bound in Equation equation 8. Since $d \le \frac{1}{C_\kappa}T^{1-\rho}$, we can set $\beta = \sqrt{C_\kappa}T^{\frac{\rho}{2}-\frac{1}{2}}$ to satisfy $\|\theta\|_2 \le 1$. Then we obtain $\mathbb{E}\left\{\sum_{t=1}^{T}\mu_\theta(a^*) - \mu_\theta(a_t)\right\} \ge dC_\kappa T^\rho/4$. Notice that the corruption level each round is bounded by $\beta\|a_t\|_\infty$. Consider the scenario when the radius of the *action* space $\mathcal{A}$ is upper bounded by $\beta$, then the corruption budget each round is bounded by $C_\kappa t^{-1+\rho}$, which satisfies the definition of $\rho$-Imperfect Human Feedback, which completes the proof. $\qquad\square$

**Lemma C.4.** *Let's consider the utility function $\mu_\theta(a) := \theta^\top a - \frac{1}{2}\|a\|_2^2$. Let $\hat{v}_1, \hat{v}_2, \ldots, \hat{v}_T$ be a sequence of* corrupted reward feedback *obtained by a learner. Then there exists a $\theta \in \mathbb{R}^d, \|\theta\|_2 \le 1$, uniformly drawn from $\{-\beta, \beta\}^d$, such that the regret suffered by the learner is*

$$\mathbb{E}\left\{ \sum_{t=1}^{T} \mu_\theta(a^*) - \mu_\theta(a_t) \right\} \ge \frac{dT\beta^2}{4}\left( 1 - \sqrt{\frac{1}{d}\sum_{i=1}^{d}\sum_{t=1}^{T} \mathbb{D}_{t,i}} \right).$$

$\mathbb{D}_{t,i} := \sup_{\{\theta_j\}_{j \ne i}} D_{KL}\left( \mathbb{P}(\hat{v}_t|\theta_i > 0, \{\theta_j\}_{j \ne i}, \{\hat{v}_l\}_{l=1}^{t-l}) \| \mathbb{P}(\hat{v}_t|\theta_i < 0, \{\theta_j\}_{j \ne i}, \{\hat{v}_l\}_{l=1}^{t-l}) \right)$. *$D_{KL}$ is the KL divergence between two distributions.*

*Proof.* Using the similar argument in Shamir (2013), assume that the learner is deterministic: $a_t$ is a deterministic function of the realized corrupted reward feedback $\hat{v}_1, \hat{v}_2, \ldots, \hat{v}_{t-1}$ at $a_1, a_2, \ldots, a_{t-1}$. This assumption is without loss of generality, since any random learners can be seen as a randomization over deterministic learning algorithms. Thus a lower bound which holds uniformly for any deterministic learner would also hold over a randomization. To lower bound equation 9, we use Lemma C.5, which relates this to the question of how informative are the query values (as measured by Kullback-Leibler divergence) for determining the sign of $\theta$'s coordinates. Intuitively, the more similar the query values are, the smaller is the KL divergence and the harder it is to distinguish the true sign of $\theta_i$, leading to a larger lower bound. In addition, we are facing a powerful adversary who

has the complete knowledge of the problem and is able to add corruption on the query value to make they are even more similar, which resulting a even smaller KL divergence, consequently, an even larger lower bound. Let $\bar{a}_T := \frac{1}{T} \sum_{t=1}^{T} a_t$ represent the average action, we have

$$
\begin{aligned}
\mathbb{E} \left\{ \sum_{t=1}^{T} \mu_\theta(a^*) - \mu_\theta(a_t) \right\} &= T\mathbb{E} \left\{ \frac{1}{T} \sum_{t=1}^{T} \mu_\theta(a^*) - \mu_\theta(a_t) \right\} \\
&\geq T\mathbb{E} \left( \mu_\theta(a^*) - \mu_\theta(\bar{a}_T) \right) \\
&\geq T\mathbb{E} \left( \frac{1}{2} \|\bar{a}_T - \theta\|^2 \right) \\
&= T\mathbb{E} \left( \frac{1}{2} \sum_{i=1}^{d} (\bar{a}_i - \theta_i)^2 \right) \\
&\geq \mathbb{E} \left( \frac{\beta^2 T}{2} \sum_{i=1}^{d} \mathbb{I}_{\bar{a}_i \theta_i < 0} \right) \\
&\geq \frac{dT\beta^2}{4} \left( 1 - \sqrt{\frac{1}{d} \sum_{i=1}^{d} \sum_{t=1}^{T} \mathbb{D}_{t,i}} \right).
\end{aligned} \tag{9}
$$

We get the second inequality by using the fact that $\mu_\theta$ is 1-strongly concave. We get the last inequality by using Lemma C.5, which completes the proof.

**Lemma C.5** (Lemma 4 in (Shamir, 2013)). *Let $\theta$ be a random vector, none of those coordinates is supported on 0. Let $\hat{v}_1, \hat{v}_2, \ldots, \hat{v}_T$ be a sequence of values obtained by a deterministic learner returning a point $\bar{a}_T$ (so that the action $a_t$ is a deterministic function of $\hat{v}_1, \ldots, \hat{v}_{t-1}$ and $\bar{a}_T$ is a deterministic function of $\hat{v}_1, \ldots, \hat{v}_T$). Then we have*

$$
\mathbb{E} \left( \sum_{i=1}^{d} \mathbb{I}_{\bar{a}_i \theta_i} \right) \geq \frac{d}{2} \left( 1 - \sqrt{\frac{1}{d} \sum_{i=1}^{d} \sum_{t=1}^{T} \mathbb{D}_{t,i}} \right),
$$

*where $\mathbb{D}_{t,i} = \sup_{\{\theta_j\}_{j \neq i}} D_{KL} \left( \mathbb{P}(\hat{v}_t | \theta_i > 0, \{\theta_j\}_{j \neq i}, \{\hat{v}_l\}_{l=1}^{t-l}) || \mathbb{P}(\hat{v}_t | \theta_i < 0, \{\theta_j\}_{j \neq i}, \{\hat{v}_l\}_{l=1}^{t-l}) \right)$, $D_{KL}$ represents the KL divergence between two distributions.*

$\square$

$\square$

## C.2 PROOF FOR PROPOSITION 1:

**Proposition** (Proposition 1 restated). *There exists an LIHF instance with $\rho$-Imperfect Human Feedback (Definition 2.1), linear user utility $\mu$, and link function $\sigma$, such that any learner has to suffer $Reg_T \geq \Omega(d \max\{\sqrt{T}, T^\rho\})$, even with the knowledge of $\rho$.*

*Proof.* The proof structure is very similar to the lower bound proof in Theorem 3.1. We start by discussing the value of $\rho$. Consider the scenario when $\rho \leq 0.5$. Applying Lemma C.6 and C.1 together with linear link function, we have $Reg_T \geq \Omega(d \max\{\sqrt{T}, T^\rho\})$, which completes the proof.

**Lemma C.6** ((Dani et al., 2008)). *Let $\mathcal{A} = [-1, 1]^d$ and $\Theta = [-T^{-\frac{1}{2}}, T^{-\frac{1}{2}}]^d$. Consider the linear reward function $r_t = \theta^\top A_t + \epsilon_t$, $\epsilon_t$ is independent standard Gaussian noise. Then for any learner, there exists a vector $\theta \in \Theta$ such that*

$$
Reg_T(\mathcal{A}, \theta) \geq \frac{\exp(-2)}{8} d\sqrt{T}.
$$

Now, let's focus on the scenario when $\rho > 0.5$ and the essence is to extend Lemma C.6 to the scenario in presence of $\rho$-Imperfect Human Feedback, which is shown in Lemma C.7

**Lemma C.7.** *Assume that the* action *space* $\mathcal{A} \subset [-1, 1]^d$. *Given* $\rho$-*Imperfect Human Feedback (Definition 2.1), for stochastic linear bandit, there exists a* $\theta \in [-C_\kappa T^{\rho-1}, C_\kappa T^{\rho-1}]^d$ *such that any learner even with the knowledge of* $\rho$ *has to suffer regret no less than* $\frac{d}{8} C_\kappa T^\rho$.

*Proof.* The proof extends Lemma C.6 to a scenario in presence of corruption. We want to construct a parameter family and a corruption strategy such that for all algorithm, it will occur at least $\Omega(dT^\rho)$ regret. Consider the action set $\mathcal{A} \in [-1, 1]^d$ and $\Theta = \{-\beta, \beta\}^d$. For any learner, we can lower bound its regret by

$$
\text{Reg}_T(\mathcal{A}, \theta) = \mathbb{E}_\theta \left[ \sum_{t=1}^T \sum_{i=1}^d (\text{sign}(\theta_i) - a_{ti})\theta_i \right]
$$

$$
\geq \beta \sum_{i=1}^d \mathbb{E}_\theta \left[ \sum_{t=1}^T \mathbb{I}\{\text{sign}(a_{ti}) \neq \text{sign}(\theta_i)\} \right]
$$

$$
\geq \frac{T\beta}{2} \sum_{i=1}^d \mathbb{P}_\theta \left( \sum_{t=1}^T \mathbb{I}\{\text{sign}(a_{ti}) \neq \text{sign}(\theta_i)\} \geq \frac{T}{2} \right)
$$

Let's denote

$$
p_{\theta_i} = \mathbb{P}_\theta \left( \sum_{t=1}^T \mathbb{I}\{\text{sign}(a_{ti}) \neq \text{sign}(\theta_i)\} \geq \frac{T}{2} \right)
$$

Let $i \in [d]$ and $\theta \in \Theta$ be fixed, and let $\theta'_j = \theta_j$, for $j \neq i$ and $\theta'_i = -\theta_i$. Then using Lemma C.8, we have

$$
p_{\theta_i} + p_{\theta'_i} \geq \frac{1}{2} \exp \left( -\mathbb{E} \left[ \sum_{t=1}^T D_{\text{KL}}(P_{a_t}, P'_{a_t}) \right] \right).
$$

$P_{a_t}$ is the distribution of corrupted reward observed by the learner after playing action $a_t$ when reward parameter is $\theta$. Similarly, $P'_{a_t}$ is the distribution of corrupted observed by the learner after playing action $a_t$ when the reward parameter is $\theta'$.

**Lemma C.8** (Bretagnolle-Huber inequality). *Let* $\mathbb{P}$ *and* $\mathbb{Q}$ *be probability measures on the same measurable space* $(\Omega, \mathcal{F})$. *Let* $A \in \mathcal{F}$ *be any arbitrary event and* $A^c$ *is the complement of A. Then we have*

$$
\mathbb{P}(A) + \mathbb{Q}(A^c) \geq \frac{1}{2} \exp \left( -D_{KL}(\mathbb{P}, \mathbb{Q}) \right). \tag{10}
$$

In presence of adversarial corruption, for $\theta$, the corrupted reward is

$$
\hat{r}_t = \sum_{j \neq i} a_{tj}\theta_j + a_{ti}\theta_i + c_t(a_t|\theta) + \epsilon_t. \tag{11}
$$

For $\theta'$, we have

$$
\hat{r}_t = \sum_{j \neq i} a_{tj}\theta_j - a_{ti}\theta_i + c_t(a_t|\theta') + \epsilon_t. \tag{12}
$$

Consider the corruption strategy such that $c_t(a_t|\theta) = -a_{t,i}\theta_i$ in equation 11 and $c_t(a_t|\theta') = a_{t,i}\theta_i$ in equation 12. This is achievable since we assume the adversary has complete knowledge of the problem instance. By doing so, the corrupted reward $\hat{r}_t$ are from the same distribution regardless whether the preference parameter is $\theta$ or $\theta'$. This is because $\theta$ and $\theta'$ only differs in the $i$-th coordinate with the magnitude $\beta$, and this difference could be masked by the corruption, resulting in $\mathbb{E} \left[ \sum_{t=1}^T D_{\text{KL}}(P_{a_t}, P'_{a_t}) \right] = 0$, which makes it indistinguishable by the learner. Notice that executing such a corruption strategy requires total corruption budget

$$
\sum_{t=1}^T |c_t(a_t)| \leq \beta \sum_{t=1}^T \|a_t\|_\infty \leq C_\kappa T^\rho. \tag{13}
$$

Choose $\beta = C_\kappa T^{\rho-1}$, Equation equation 13 holds. Therefore, we have

$$
p_{\theta_i} + p_{\theta'_i} \geq \frac{1}{2}.
$$

Applying an "averaging hammer" over all $\theta \in \Theta$, which satisfies $|\Theta| = 2^d$, we get

$$\sum_{\theta \in \Theta} \frac{1}{|\Theta|} \sum_{i=1}^{d} p_{\theta_i} = \frac{1}{|\Theta|} \sum_{i=1}^{d} \sum_{\theta \in \Theta} p_{\theta_i} \geq \frac{d}{4}.$$

This implies that there exists a $\theta \in \Theta$ such that $\sum_{i=1}^{d} p_{\theta_i} \geq \frac{d}{4}$. Therefore we have

$$\text{Reg}_T(\mathcal{A}, \theta) \geq \frac{d C_\kappa}{8} T^\rho,$$

which completes the proof. We want to highlight that the corruption level each round $|c_t(a_t)|$ is bounded by $\beta \|a_t\|_\infty \leq C_\kappa T^{\rho-1} \leq C_\kappa t^{\rho-1}, \forall t$, satisfying definition of $\rho$-Imperfect Human Feedback. $\qquad\square$

$\hfill\square$

## D  PROOF FOR THEOREM 4.1

**Theorem** (Theorem 4.1 restated). *RoSMID satisfies $Reg_T \leq \tilde{\mathcal{O}}(d \max\{\sqrt{T}, T^\rho\})$ for any $\rho$-LIHF problem.*

*Proof.* At the beginning of the proof, we formally define self-concordance (see Definition D.1). The input, $\nu$-self-concordant function $\mathcal{R}$, serves as a regularizor in RoSMID (see Algorithm 1).

**Definition D.1.** *(Self-concordance.) A function $\mathcal{R} : int(\mathcal{A}) \to \mathbb{R}$ is self-concordant if it satisfies*

1. *$\mathcal{R}$ is three times continuously differentiable, convex, and approaches infinity along any sequence of points approaching the boundary of $int(\mathcal{A})$.*

2. *For every $h \in \mathbb{R}^d$ and $x \in int(\mathcal{A})$, $|\nabla^3 \mathcal{R}(x)[h,h,h]| \leq 2 \left( h^\top \nabla^2 \mathcal{R}(x) h \right)^{\frac{3}{2}}$ holds, where $|\nabla^3 \mathcal{R}(x)[h,h,h]| := \frac{\partial^3 \mathcal{R}}{\partial t_1 \partial t_2 \partial t_3}(x + t_1 h + t_2 h + t_3 h)|_{t_1=t_2=t_3=0}$.*

*In addition to these two conditions, if for every $h \in \mathbb{R}^d$ and $x \in int(\mathcal{A})$, $|\nabla \mathcal{R}(x)^\top h \leq \nu^{\frac{1}{2}} (h^\top \nabla^2 \mathcal{R}(x) h)^{\frac{1}{2}}|$ for a positive real number $\nu$, $\mathcal{R}$ is $\nu$-self-concordant.*

Additionally, we remind the reader of the standard assumptions (Yue & Joachims, 2009; Kumagai, 2017) used in our analysis.

1. the action set $\mathcal{A}$ is a convex compact set that contains the origin, has non-empty interior, and is contained in a $d$-dimensional ball of radius $R$;

2. the utility function $\mu : \mathcal{A} \to \mathbb{R}$ is strongly concave and twice-differentiable. The following constants are useful for our algorithm analysis: $\mu$ is $L^\mu$-Lipschitz, $\alpha$-strongly concave, $\kappa$-smooth, bounded $R^\mu := \sup_{a,a'} \mu(a) - \mu(a')$, and $a^* := \arg\max_{a \in \mathcal{A}} \mu(a)$ is the unique optimal within $\mathcal{A}$;

3. the link function $\sigma : \mathbb{R} \to [0,1]$ is smooth, rotation-symmetric (i.e. $\sigma(x) = 1 - \sigma(-x)$), and concave for any $x \geq 0$. For the ease of analysis: let $l_1^\sigma$ [resp. $L_1^\sigma$] denote the lower [resp. upper] bound of the first-order derivative of $\sigma$. $\sigma$ is $L^\sigma$-Lipschitz and its second-order derivative $\sigma''$ is $L_2^\sigma$-Lipschitz and upper bounded by $R_2^\sigma$.

Under these assumption, Lemma D.1 implies $\text{Reg}_T$ defined in equation 1 and $\text{Reg}_T^{\text{FO}}$ defined in equation 3 has the same order.

**Lemma D.1** (Lemma 12 in Kumagai (2017)). *With $Reg_T^{FO}$ defined in equation 3, we have*

$$\frac{Reg_T}{L_1^\sigma} \leq Reg_T^{FO} \leq \frac{Reg_T}{l_1^\sigma}. \tag{14}$$

In the following, we show the proof. The main proof can be divided into four key steps. Step 1 is natural, which quantifies the bias in gradient estimation by RoSMID due imperfect feedback from corrupted utilities. Building on this, Step 2 develops a regret decomposition lemma for dueling bandits under corruption, decomposing the regret into two components: regret from sub-optimal decisions and regret from feedback error. Techniques from previous analyses of continuous dueling bandits handle the sub-optimal decision regret, thus in Step 3, we focus on deriving new techniques to bound the regret from feedback error, a unique challenge in our setting. We identify a tight upper bound for this error. Finally, Step 4, the only step relying on the decaying corruption level in $\rho$-imperfect human feedback, uses this assumption, along with the carefully chosen learning rate $\eta_\rho$ and an iterative regret refinement process, to ensure the tightness of the analysis in Step 3, thus completing the proof.

**Step 1: quantifying bias of gradient estimation in $\rho$-LIHF**

Our proof starts from quantifying the bias of the gradient $\hat{g}_t$ caused by $\rho$-imperfect human feedback, as shown in the following Lemma 4.1.

**Lemma** (Lemma 4.1 restated). *The gradient $\hat{g}_t$ in Line 6 of Algorithm 1 satisfies*

$$\mathbb{E}\left(\hat{g}_t | a_t\right) = \nabla|_{a=a_t} \bar{P}_t(a) - \mathbb{E}\left(b_t | a_t\right).$$

*where $\bar{P}_t(a) := \mathbb{E}_{x \in \mathbb{B}}\left[\mathbb{P}(a_t \succ a + \nabla^2 R_t(a_t)^{-\frac{1}{2}} x)\right]$ (with $\mathbb{B}$ as the unit ball) is the smoothed probability that $a_t$ is preferred over any action $a$, and $b_t = d\left(\mathbb{P}(a_t \succ a'_t) - \hat{\mathbb{P}}(a_t \succ a'_t)\right) \nabla^2 \mathcal{R}_t(a_t)^{\frac{1}{2}} u_t$.*

*Proof.* If we do not have corruption (i.e. the probability for observing noisy comparative feedback $\mathcal{F}(a, a') = 1$ is based on true cost difference $\mu(a) - \mu(a')$), then the uncorrupted gradient $g_t := \mathcal{F}(a'_t, a_t) d\nabla^2 \mathcal{R}_t(a_t)^{1/2} u_t$ should be an unbiased estimate of $\nabla \bar{P}_t(a_t)$, i.e.

$$\mathbb{E}(g_t | a_t) = \nabla \bar{P}_t(a_t).$$

We use $\nabla \bar{P}_t(a_t)$ as a shorthand of $\nabla|_{a=a_t} \bar{P}_t(a)$. We use $P_t(a)$ as a shorthand of $P_t(a) := \sigma(\mu(a_t) - \mu(a))$ equivalent to $\mathbb{P}(a_t \succ a)$, and $\hat{P}_t(a) := \sigma(\mu(a_t) - \mu(a) + c_t(a_t, a'_t))$, equivalent to $\hat{\mathbb{P}}(a_t \succ a)$ under our modelling assumption. The proof is similar to Lemma 2.1 in Flaxman et al. (2005). Using the Law of total expectation, we have

$$\begin{aligned}
\mathbb{E}(g_t | a_t) &= \mathbb{E}_{u_t}[\mathbb{E}(g_t | a_t, u_t)] \\
&= \mathbb{E}_{u_t}\left(d\mathbb{E}(P_t(a_t + \nabla^2 \mathcal{R}_t(a_t)^{-\frac{1}{2}} u_t)\nabla^2 \mathcal{R}_t(a_t)^{\frac{1}{2}} u_t | a_t, u_t)\right) \\
&= d\mathbb{E}(P_t(a_t + \nabla^2 \mathcal{R}_t(a_t)^{-\frac{1}{2}} u_t)\nabla^2 \mathcal{R}_t(a_t)^{\frac{1}{2}} u_t | a_t) \\
&= \nabla \mathbb{E}_{x \in \mathbb{B}}\left(P_t(a_t + \nabla^2 \mathcal{R}_t(a_t)^{-\frac{1}{2}} x | a_t)\right) \\
&= \nabla \bar{P}_t(a_t).
\end{aligned}$$

We get the second inequality by using the definition of $\mathcal{F}(a'_t, a_t)$. We use the Stroke's Theorem to get the second last equality. The gradient which we get to perform gradient descent $\hat{g}_t$ is corrupted. If we let $a'_t = a_t + \nabla^2 \mathcal{R}_t(a_t)^{-\frac{1}{2}} u_t$, we have

$$\begin{aligned}
\mathbb{E}(\hat{g}_t | a_t) &= \mathbb{E}_{u_t}(\mathbb{E}(\hat{g}_t | a_t, u_t)) \\
&= \mathbb{E}_{u_t}\left(d\mathbb{E}(\hat{P}_t(a'_t)\nabla^2 \mathcal{R}_t(a_t)^{\frac{1}{2}} u_t | a_t, u_t)\right) \\
&= d\mathbb{E}\left(\hat{P}_t(a'_t)\nabla^2 \mathcal{R}_t(a_t)^{\frac{1}{2}} u_t | a_t\right) \\
&= d\mathbb{E}\left(\left[P_t(a'_t) + \hat{P}_t(a'_t) - P_t(a'_t)\right]\nabla^2 R_t(a_t)^{\frac{1}{2}} u_t | a_t\right) \\
&= \nabla \mathbb{E}_{x \in \mathbb{B}}\left(P_t(a_t + \nabla^2 \mathcal{R}_t(a_t)^{-\frac{1}{2}} x | a_t)\right) + d\mathbb{E}\left[\left(\hat{P}_t(a'_t) - P_t(a'_t)\right)\nabla^2 \mathcal{R}_t(a_t)^{\frac{1}{2}} u_t | a_t\right] \\
&= \nabla \bar{P}_t(a_t) + d\mathbb{E}\left[\left(\hat{P}_t(a'_t) - P_t(a'_t)\right)\nabla^2 \mathcal{R}_t(a_t)^{\frac{1}{2}} u_t | a_t\right] \\
&= \mathbb{E}(g_t | a_t) + d\mathbb{E}\left[\left(\hat{P}_t(a'_t) - P_t(a'_t)\right)\nabla^2 \mathcal{R}_t(a_t)^{\frac{1}{2}} u_t | a_t\right].
\end{aligned}$$

We get the second last equality by using the fact that $g_t$ is unbiased, which we proved earlier. If we defined $b_t$ as

$$b_t := d\left(P_t(a_t + \nabla^2 \mathcal{R}_t(a_t)^{-\frac{1}{2}} u_t) - \hat{P}_t(a_t + \nabla^2 \mathcal{R}_t(a_t)^{-\frac{1}{2}} u_t)\right) \nabla^2 \mathcal{R}_t(a_t)^{\frac{1}{2}} u_t,$$

we get

$$\mathbb{E}(g_t | a_t) = \mathbb{E}(\hat{g}_t | a_t) + \mathbb{E}\left[b_t | a_t\right].$$

$\square$

**Step 2: decomposing regret for dueling bandits into regret of decision and feedback error**

Core to our proof is a "regret decomposion" lemma below that upper bounds the regret by two parts: regret of sub-optimal decision making under uncertainty, referred as regret of decision (first term in equation 2), and feedback error (second term in equation 2).

**Lemma** (Lemma 4.2 restated). *The $Reg_T$ of Algorithm 1 satisfies:*

$$Reg_T \leq \underbrace{\frac{C_0}{\eta_\rho} \log(T) + 8d^2 \eta_\rho T + 2L^\mu L_1^\sigma R}_{Regret\ of\ Decision} + \underbrace{2\mathbb{E}\left\{\sum_{t=1}^{T} b_t^\top (a_t - a_T^*)\right\}}_{Feedback\ Error},$$

*where $C_0 := (2\nu + 4L_1^\sigma \kappa + 4R_2^\sigma (L^\mu)^2 + L^\mu L_1^\sigma \kappa)/\lambda$, and $a_T^* := \frac{1}{T}a_1 + (1 - \frac{1}{T})a^*$.*

*Proof.* The cornerstone of the analysis below is to separate the impact of $b_t$ from the total regret.

$$\text{Reg}_T \leq 2\mathbb{E}\left[\sum_{t=1}^{T}(P_t(a_t) - P_t(a_T^*))\right] + \frac{L^\mu L_1^\sigma \kappa}{\lambda \eta_\rho} + 2L^\mu L_1^\sigma R$$

$$\leq 2\left(\mathbb{E}\left\{\sum_{t=1}^{T}(\bar{P}_t(a_t) - \bar{P}_t(a_T^*))\right\} + \mathbb{E}\left\{\sum_{t=1}^{T}(P_t(a_t) - \bar{P}_t(a_t))\right\} + \mathbb{E}\left\{\sum_{t=1}^{T}(\bar{P}_t(a_T^*) - P_t(a_T^*))\right\}\right) + \frac{L^\mu L_1^\sigma \kappa}{\lambda \eta_\rho} + 2L^\mu L_1^\sigma R$$

$$\leq 2\mathbb{E}\left\{\sum_{t=1}^{T}(\bar{P}_t(a_t) - \bar{P}_t(a_T^*))\right\} + \frac{4L_1^\sigma \kappa + 4R_2^\sigma (L^\mu)^2 + L^\mu L_1^\sigma \kappa}{\lambda \eta_\rho} \log T + 2L^\mu L_1^\sigma R.$$

We get the first inequality by using Lemma D.2. We get the last inequality by because

$$\bar{P}_t(a) - P_t(a) \leq \frac{L_1^\sigma \kappa + R_2^\sigma (L^\mu)^2}{2} \|\nabla^2 \mathcal{R}_t(a_t)^{-\frac{1}{2}} u_t\|^2 \leq \frac{L_1^\sigma \kappa + R_2^\sigma (L^\mu)^2}{\lambda \eta_\rho t},$$

and

$$\mathbb{E}\left\{\sum_{t=1}^{T}(P_t(a_t) - P_t(a_T^*))\right\} \leq \mathbb{E}\left\{\sum_{t=1}^{T}(\bar{P}_t(a_t) - \bar{P}_t(a_T^*))\right\} + 2\frac{L_1^\sigma \kappa + R_2^\sigma (L^\mu)^2}{\lambda \eta_\rho} \log T.$$

**Lemma D.2** (Lemma 5 in Kumagai (2017)).

$$Reg_T \leq 2\mathbb{E}\left[\sum_{t=1}^{T}(P_t(a_t) - P_t(a_T^*))\right] + \frac{L^\mu L_1^\sigma \kappa}{\lambda \eta_\rho} + 2L^\mu L_1^\sigma R.$$

Then it remains to bound $\mathbb{E}\{\sum_{t=1}^{T} \bar{P}_t(a_t) - \bar{P}_t(a_T^*)\}$. And we have the following.

$$\mathbb{E}\{\bar{P}_t(a_t) - \bar{P}_t(a_T^*)\} \leq \mathbb{E}\left[\nabla \hat{P}_t^\top (a_t)(a_t - a_T^*) - \frac{L_1^\sigma \alpha}{4} \|a_t - a_T^*\|^2\right]$$

$$= \mathbb{E}\left[g_t^\top (a_t - a_T^*) - \frac{L_1^\sigma \alpha}{4} \|a_t - a_T^*\|^2\right]$$

$$= \mathbb{E}\left[(\hat{g}_t + b_t)^\top (a_t - a_T^*) - \frac{L_1^\sigma \alpha}{4} \|a_t - a_T^*\|^2\right]$$

$$= \mathbb{E}\left[\hat{g}_t^\top (a_t - a_T^*) - \frac{L_1^\sigma \alpha}{4} \|a_t - a_T^*\|^2\right] + \mathbb{E}\{b_t^\top (a_t - a_T^*)\},$$

where the first and second equality is resulted from Lemma 4.1. Using the definition of $a_{t+1}$ in Algorithm 1, we have

$$\nabla \mathcal{R}_t(a_{t+1}) - \nabla \mathcal{R}_t(a_t) = \eta_\rho \hat{g}_t.$$

Therefore we have

$$\mathbb{E}\left[\hat{g}_t^\top(a_t - a_T^*) - \frac{L_1^\sigma \alpha}{4}\|a_t - a_T^*\|^2\right] = \frac{1}{\eta_\rho}\mathbb{E}\left[(\nabla \mathcal{R}_t(a_{t+1}) - \nabla \mathcal{R}_t(a_t))^\top(a_t - a_T^*) - \frac{L_1^\sigma \alpha \eta_\rho}{4}\|a_t - a_T^*\|^2\right]$$

$$= \frac{1}{\eta_\rho}\mathbb{E}\left[D_{\mathcal{R}_t}(a_T^*, a_t) + D_{\mathcal{R}}(a_t, a_{t+1}) - D_{\mathcal{R}_t}(a_T^*, a_{t+1}) - \frac{L_1^\sigma \alpha \eta_\rho}{4}\|a_t - a_T^*\|^2\right].$$

$D_{\mathcal{R}}(a, b)$ is the Bregman divergence associated with $\mathcal{R}$, defined by

$$D_{\mathcal{R}}(a, b) = \mathcal{R}(a) - \mathcal{R}(b) - \nabla\mathcal{R}(b)^\top(a - b).$$

Then we can get equation 15 by using Lemma D.3, D.4, D.5.

$$\mathbb{E}\left\{\sum_{t=1}^T (\bar{P}_t(a_t) - \bar{P}_t(a_T^*))\right\} \leq \frac{\nu \log(T)}{\eta_\rho} + 4d^2\eta_\rho T + \mathbb{E}\{\sum_{t=1}^T b_t^\top(a_t - a_T^*)\}. \qquad (15)$$

**Lemma D.3** (Lemma 10 in Kumagai (2017)). *Let $\mathcal{R}_t^*(a) = \sup_{x \in \mathbb{R}^d} x^\top a - \mathcal{R}_t(a)$ denote the Frenchel dual of $\mathcal{R}_t$. Then we have*

$$\sum_{t=1}^T \mathbb{E}\left[\hat{g}_t^\top(a_t - a_T^*) - \frac{L_1^\sigma \alpha}{4}\|a_t - a_T^*\|^2\right] \leq \frac{1}{\eta_\rho}\left(\mathcal{R}(a_T^*) - \mathcal{R}(a_1) + \mathbb{E}\left[\sum_{t=1}^T D_{\mathcal{R}_t^*}(\nabla\mathcal{R}_t(a_t) - \eta_\rho\hat{g}_t, \nabla\mathcal{R}_t(a_t))\right]\right).$$

**Lemma D.4** (Lemma 11 in Kumagai (2017)). *When $\eta_\rho \leq \frac{1}{2d}$, we have*

$$D_{\mathcal{R}_t^*}(\nabla\mathcal{R}_t(a_t) - \eta_\rho\hat{g}_t, \nabla\mathcal{R}_t(a_t)) \leq 4d^2\eta_\rho^2.$$

**Lemma D.5** (Lemma 4 in Hazan & Levy (2014)). $\mathcal{R}(a_T^*) - \mathcal{R}(a_1) \leq \nu \log(T)$.

If we let $C_0 := (2\nu + 4L_1^\sigma\kappa + 4R_2^\sigma(L^\mu)^2 + L^\mu L_1^\sigma\kappa)/\lambda$, we have

$$\text{Reg}_T \leq \frac{2\nu \log(T)}{\eta_\rho} + 8d^2\eta_\rho T + 2\mathbb{E}\left\{\sum_{t=1}^T b_t^\top(a_t - a_T^*)\right\} + \frac{4L_1^\sigma\kappa + 4R_2^\sigma(L^\mu)^2 + L^\mu L_1^\sigma\kappa}{\lambda\eta_\rho}\log T + 2L^\mu L_1^\sigma R$$

$$= \underbrace{\frac{C_0}{\eta_\rho}\log(T) + 8d^2\eta_\rho T + 2L^\mu L_1^\sigma R}_{\textit{Regret of Decision}} + \underbrace{2\mathbb{E}\left\{\sum_{t=1}^T b_t^\top(a_t - a_T^*)\right\}}_{\textit{Feedback Error}},$$

which completes the proof. $\qquad\qquad\square$

**Step 3: upper bounding the regret from feedback error**

Bounding the regret of decision term in equation 2 is standard. Thus, this step develops a novel upper bound for bounding the regret from feedback error, as shown below.

**Lemma** (Lemma 4.3 restated). *The feedback error term in equation 2 can be bounded as follows:*

$$\mathbb{E}\left\{\sum_{t=1}^T b_t^\top(a_t - a_T^*)\right\} \leq C_1 d\sqrt{\eta_\rho}\mathbb{E}\left\{\sum_{t=1}^T \sqrt{t}|c_t(a_t, a_t')|\sqrt{\mu(a^*) - \mu(a_t)}\right\} + C_2 dT^\rho + 2d\sqrt{\lambda}\sqrt{\eta_\rho T},$$

*where $C_1 := \sqrt{\frac{2\lambda(L^\sigma)^2}{\alpha}}$, and $C_2 := 2L^\sigma R\sqrt{\sup_{a \in \mathcal{A}} \lambda_{\max}(\nabla^2\mathcal{R}(a)) + 2\phi}$.*

*Proof.* In the following, we will analyze the expected cumulative sum of the bias $b_t$ on the total regret, which is $\mathbb{E}\left\{\sum_{t=1}^T b_t^\top(a_t - a_T^*)\right\}$. By Cauchy-Schwartz inequality, $b_t^\top(a_t - a_T^*) \leq \|b_t\|_2\|(a_t - a_T^*)\|_2$. We can bound the $\|b_t\|_2$ by the following.

$$b_t^\top b_t = d^2\left(P_t(a_t + \nabla^2\mathcal{R}_t(a_t)^{-\frac{1}{2}}u_t) - \hat{P}_t(a_t + \nabla^2\mathcal{R}_t(a_t)^{-\frac{1}{2}}u_t)\right)^2 u_t^\top\nabla^2\mathcal{R}_t(a_t)u_t$$

$$\leq \left(P_t(a_t + \nabla^2\mathcal{R}_t(a_t)^{-\frac{1}{2}}u_t) - \hat{P}_t(a_t + \nabla^2\mathcal{R}_t(a_t)^{-\frac{1}{2}}u_t)\right)^2 d^2\lambda_{\max}(\nabla^2\mathcal{R}_t(a_t)).$$

We get the first equality by Lemma 4.1. We get the inequality by using the fact $\|u_t\|_2 = 1$. If we let $a'_t = a_t + \nabla^2 \mathcal{R}_t(a_t)^{-\frac{1}{2}} u_t$, then we have

$$|P_t(a'_t) - \hat{P}_t(a'_t)| = |\sigma(f(a_t) - f(a'_t)) - \sigma(f(a_t) - f(a'_t) + c_t(a_t, a'_t))|$$
$$\leq \min(2, L^\sigma |c_t(a_t, a'_t)|).$$

We get the first equality by using the definition of $P_t(a_t)$ and $\hat{P}_t(a_t)$. We get the first inequality by using the Lipschitz property of $\sigma$. Therefore, we have

$$\|b_t\|_2 \leq d\sqrt{\lambda_{\max}(\nabla^2 \mathcal{R}_t(a_t))} \min\{2, L^\sigma |c_t(a_t, a'_t)|\}.$$

If we use $\lambda_R^* := \sup_{a \in \mathcal{A}} \lambda_{\max}(\nabla^2 \mathcal{R}(a))$, then we have

$$\lambda_{\max}(\nabla^2 \mathcal{R}_t(a_t)) = \lambda_{\max}(\nabla^2 \mathcal{R}(a_t)) + \lambda \eta_\rho t + 2\phi \leq \lambda_R^* + 2\phi + \lambda \eta_\rho t.$$

Therefore we have

$$\mathbb{E}\left\{\sum_{t=1}^T b_t^\top (a_t - a_T^*)\right\} \leq \mathbb{E}\left\{\sum_{t=1}^T \|b_t\|_2 \|a_t - a_T^*\|_2\right\}$$

$$\leq \mathbb{E}\left\{\sum_{t=1}^T (d\sqrt{\lambda_{\max}(\nabla^2 \mathcal{R}_t(a_t))} \min(2, L^\sigma |c_t(a_t, a'_t)|)) \|a_t - a_T^*\|_2\right\}$$

$$\leq \mathbb{E}\left\{\sum_{t=1}^T (d\sqrt{\lambda_R^* + 2\phi + \lambda \eta_\rho t} \min(2, L^\sigma |c_t(a_t, a'_t)|)) \|a_t - a_T^*\|_2\right\}$$

$$\leq \mathbb{E}\left\{\sum_{t=1}^T (d\sqrt{\lambda_R^* + 2\phi} \min(2, L^\sigma |c_t(a_t, a'_t)|) \|a_t - a_T^*\|_2\right\} + \mathbb{E}\left\{\sum_{t=1}^T (d\sqrt{\lambda \eta_\rho t} \min(2, L^\sigma |c_t(a_t, a'_t)|) \|a_t - a_T^*\|_2\right\}$$

$$\leq 2RdL^\sigma \sqrt{\lambda_R^* + 2\phi} T^\rho + d\sqrt{\lambda \eta_\rho} \mathbb{E}\left\{\sum_{t=1}^T \sqrt{t} \min(2, L^\sigma |c_t(a_t, a'_t)|) \|a_t - a_T^*\|_2\right\} \tag{16}$$

$$\leq 2RdL^\sigma \sqrt{\lambda_R^* + 2\phi} T^\rho + 2d\sqrt{\lambda \eta_\rho T} + d\sqrt{\lambda \eta_\rho} \mathbb{E}\left\{\sum_{t=1}^T \sqrt{t} \min(2, L^\sigma |c_t(a_t, a'_t)|) \|a_t - a^*\|_2\right\} \tag{17}$$

$$\leq 2RdL^\sigma \sqrt{\lambda_R^* + 2\phi} T^\rho + 2d\sqrt{\lambda \eta_\rho T} + d\sqrt{\lambda \eta_\rho} \mathbb{E}\left\{\sum_{t=1}^T \sqrt{t} \min(2, L^\sigma |c_t(a_t, a'_t)|) \min\left(2R, \sqrt{\frac{2}{\alpha}(\mu(a^*) - \mu(a_t))}\right)\right\}. \tag{18}$$

We get equation 16 by using the fact that $\|a_t - a_T^*\|_2 \leq 2R$, $\sum_{t=1}^T |c_t(a_t, a'_t)| \leq T^\rho$. We get equation 17 the definition of $a_T^*$, and the fact $\sqrt{a+b} \leq \sqrt{a} + \sqrt{b}$. Using the $\alpha$-strong convexity of $\mu$, we have $\|a_t - a_*\|_2 \leq \min\left(2R, \sqrt{\frac{2}{\alpha}(\mu(a^*) - \mu(a_t))}\right)$ we get the last inequality, which completes the proof. $\qquad\square$

**Step 4: The *iterative regret refinement* analysis and resultant optimal learning rate**

**Lemma** (Lemma 4.4 restated). *If $A = \mathbb{E}\left(\sum_{t=1}^T [\mu(a^*) - \mu(a_t)]\right) \leq \mathcal{O}(T^{\frac{1}{2}+\psi})$ for some $\psi \in [0, \frac{1}{2})$, then we must have $B = \mathbb{E}\left(\sum_{t=1}^T t^{\rho-\frac{1}{2}} \sqrt{\mu(a^*) - \mu(a_t)}\right) \leq \mathcal{O}(T^{-\frac{1}{4}+\frac{\psi}{2}+\rho})$.*

*Proof.* To prove Lemma 4.4, it is equivalent to prove the following Lemma D.6. If Lemma D.6 holds, let $n_t = \sqrt{\mathbb{E}(\mu(a^*) - \mu(a_t))}$, $m_t = \min(2, L^\sigma |c_t(a_t, a'_t)|)$, proves Lemma 4.4.

**Lemma D.6.** *Consider $\sum_{t=1}^T n_t^2 \leq C' T^{\frac{1}{2}+\alpha}$, with constants $\alpha \geq 0$, $C' > 0$. In addition, $n_t$ is uniformly upper bounded by a constant $K$. Specifically, we have $0 \leq n_t \leq K, \forall t$. $m_t \leq c_k t^{\rho-1}$ with constant $c_k \geq 0$. Then we have $\sum_{t=1}^T \sqrt{t} m_t n_t \leq 5\sqrt{C'} c_k T^{-\frac{1}{4}+\frac{\alpha}{2}+\rho}$.*

*Proof.* Using Abel's equality (Lemma D.7), we have

$$\sum_{t=1}^{T} \sqrt{t} m_t n_t \leq \sqrt{T} \left( \sum_{t=1}^{T} m_t n_t \right)$$

Since we have $\sum_{t=1}^{T} n_t^2 \leq C'T^{\frac{1}{2}+\alpha}$ and $0 \leq n_t \leq K, \forall t$, we have $\sum_{t=1}^{T} n_t \leq \sqrt{C'}T^{\frac{3}{4}+\frac{\alpha}{2}}, \forall t$. Moreover, for $t \geq 1$, $t^{-1+\rho} - (t+1)^{-1+\rho} \leq t^{\rho-2}$ holds for all $t \geq 1$. This is because

$$f(t) = t^{-1+\rho} \left( (1+\frac{1}{t})^{-1+\rho} + \frac{1}{t} - 1 \right).$$

is decreasing over $t \geq 1$ and $\lim_{t\to\infty} f(t) = 0$. Consequently, applying Abel's Summation Equation again, we have

$$\begin{aligned}
\sum_{t=1}^{T} m_t n_t &= \left( \sum_{t=1}^{T} n_t \right) m_T + \sum_{t=1}^{T-1} \left( \sum_{i=1}^{t} n_i \right) (m_t - m_{t+1}) \\
&\leq \sqrt{C'} c_k T^{\frac{3}{4}+\frac{\alpha}{2}} T^{-1+\rho} + \sqrt{C'} c_k \sum_{t=1}^{T-1} \left( \sum_{i=1}^{t} n_i \right) \left( t^{-1+\rho} - (t+1)^{-1+\rho} \right) \\
&\leq \sqrt{C'} c_k T^{\frac{3}{4}+\frac{\alpha}{2}} T^{-1+\rho} + \sqrt{C'} c_k \sum_{t=1}^{T-1} \left( \sum_{i=1}^{t} n_i \right) t^{-2+\rho} \\
&\leq \sqrt{C'} c_k T^{\frac{3}{4}+\frac{\alpha}{2}} T^{-1+\rho} + \sqrt{C'} c_k \sum_{t=1}^{T-1} t^{\frac{3}{4}+\frac{\alpha}{2}} t^{-2+\rho} \\
&\leq 5\sqrt{C'} c_k T^{-\frac{1}{4}+\frac{\alpha}{2}+\rho}.
\end{aligned}$$

Since $0 \leq n_t \leq K$, $\sum_{i=1}^{t} n_i$ is increasing and the increasing rate is upper bound by $t$. Together with the constraint $\sum_{t=1}^{T} n_t \leq \sqrt{C'}T^{\frac{3}{4}+\frac{\alpha}{2}}$, $\sum_{t=1}^{T-1} \left( \sum_{i=1}^{t} n_i \right) t^{-\frac{1}{2}+\rho}$ is optimized when $\sum_{i=1}^{t} n_i$ increasing in the speed of $\sqrt{C'}t^{\frac{3}{4}+\frac{\alpha}{2}}$. Because of this, the second last inequality holds.

**Lemma D.7.** *(Abel's Summation Equation (Williams, 1963)). For any numbers $a_k$, $b_k$, we have*

$$\sum_{k=1}^{n} a_k b_k = \left( \sum_{k=1}^{n} b_k \right) a_n + \sum_{k=1}^{n-1} \left( \sum_{i=1}^{k} b_i \right) (a_k - a_{k+1}).$$

□

□

After establishing Lemma 4.4, we can use it to prove the induction claim (Lemma 4.5). The challenge in this proof lies in identifying the constant $K$, which must hold across all iterations of the analysis to ensure the bound remains stable and does not diverge. This requires careful refinement of the analysis and calculations.

**Lemma** (Lemma 4.5 restated). *Consider $T > \sqrt{2L^\mu L_1^\sigma R}$ and $\eta_\rho = \frac{\sqrt{\log T}}{dT^{\max\{1/2,\rho\}}}$. If $Reg_T \leq 144dK\sqrt{\log T}T^{\rho+\frac{\frac{3}{2}-\rho}{2^k}}$ for some integer $k$, then we must also have $Reg_T \leq 144dK\sqrt{\log T}T^{\rho+\frac{\frac{3}{2}-\rho}{2^{k+1}}}$. $K$ is a instance-dependent constant, where $K := \max\left\{ C_0, \frac{C_2 C_\kappa}{\sqrt{\log T}}, (L^\sigma T_\kappa^\rho)^2, 2\sqrt{\lambda}RC_\kappa L^\sigma \right\}$, $C_0$ and $C_2$ defined in Lemma 4.2 and 4.3 respectively,*

*Proof.* In the following, we denote

$$m_t := \min\left(2, L^\sigma |c_t(a_t, a_t')|\right), n_t := \min\left(2R, \sqrt{\frac{2}{\alpha}(\mu(a^*) - \mu(a_t))}\right)$$

Continue from equation 18, we have the following.

**The Base Case:** When $k = 1$, let $C_3 = C_2 dT^\rho + 2d\sqrt{\lambda\eta_\rho T}$, we have

$$\text{Reg}_T \leq \frac{C_0}{\eta_\rho}\log(T) + 8d^2\eta_\rho T + 2L^\mu L_1^\sigma R + 2C_3 + 2d\sqrt{\lambda\eta_\rho}\mathbb{E}\left(\sum_{t=1}^T \sqrt{t}m_t n_t\right). \tag{19}$$

$$\leq \frac{C_0}{\eta_\rho}\log(T) + 8d^2\eta_\rho T + 2L^\mu L_1^\sigma R + 2C_3 + 4dR\sqrt{\lambda\eta_\rho}\mathbb{E}\left\{\sum_{t=1}^T \sqrt{t}m_t\right\} \tag{20}$$

$$\leq \frac{C_0}{\eta_\rho}\log(T) + 8d^2\eta_\rho T + 2L^\mu L_1^\sigma R + 2C_3 + 4dR\sqrt{\lambda\eta_\rho T}\left(\sum_{t=1}^T m_t\right) \tag{21}$$

$$\leq \frac{C_0}{\eta_\rho}\log(T) + 8d^2\eta_\rho T + 2L^\mu L_1^\sigma R + 2C_3 + 4dL^\sigma R\sqrt{\lambda\eta_\rho}C_\kappa T^\rho. \tag{22}$$

We get equation 19 according to Lemma 4.2 and 4.3. We get equation 20 because of $\mathbb{E}(n_t) \leq 2R$. We get equation 21 by Abel's Summation Equation (see Lemma D.7). We get equation 22 because of corruption budget. Choosing the learning rate $\eta_\rho$ according to Theorem 4.1, we obtain $\text{Reg}_T \leq 144\sqrt{\log T}KdT^{\frac{\rho+1}{2}}$. This confirms the validity of the claim when $k = 1$.

**The Induction Argument:** Let's assume that the claim holds true for a general step $k$. In the following, we will demonstrate that the claim also holds true for step $k + 1$. Because of Equation equation 14 and the induction claim, we obtain $\text{Reg}_T^{\text{FO}} \leq \frac{144}{l_1^\sigma}dK\sqrt{\log T}T^{\rho + \frac{3/2-\rho}{2^k}}$. This suggests that

$$\sum_{t=1}^T \mathbb{E}\left(\mu(a^*) - \mu(a_t)\right) \leq \frac{144}{l_1^\sigma}dK\sqrt{\log T}T^{\rho + \frac{3/2-\rho}{2^k}}.$$

By Jensen's inequality, we have

$$\mathbb{E}(n_t) = \min\left\{2, \sqrt{\frac{2}{\alpha}}\mathbb{E}\left(\sqrt{\mu(a^*) - \mu(a_t)}\right)\right\} \leq \sqrt{\frac{2}{\alpha}}\sqrt{\mathbb{E}\left(\mu(a^*) - \mu(a_t)\right)}.$$

Therefore we have

$$\mathbb{E}\left(\sum_{t=1}^T \sqrt{t}m_t n_t\right) \leq \sqrt{\frac{2}{\alpha}}\sum_{t=1}^T \sqrt{t}m_t\sqrt{\mathbb{E}\left(\mu(a^*) - \mu(a_t)\right)}.$$

By the definition of $\rho$-Imperfect Human Feedback, we have $|c_t(a_t, a_t')|$ is upper bounded by $C_k t^{-1+\rho}$.

Therefore, using Lemma 4.4 we can upper bound $\sqrt{\lambda}\sum_{t=1}^T \sqrt{t}m_t n_t$ by

$$\sqrt{\lambda}\sum_{t=1}^T \sqrt{t}m_t n_t \leq 60L^\sigma\sqrt{\frac{2\lambda}{\alpha l_1^\sigma}}(\log T)^{0.25}C_\kappa\sqrt{dK}T^{\frac{3}{2}\rho + \frac{3/2-\rho}{2^{k+1}}}$$

Since $\lambda \leq \alpha l_1^\sigma/2$, choosing $\eta_\rho = \frac{1}{d}\frac{\sqrt{\log T}}{T^\rho}$, we have

$$\text{Reg}_T \leq 2d\left(C_0 + \frac{C_2}{\sqrt{\log T}} + 60C_\kappa\sqrt{KT}^{\frac{3/2-\rho}{2^{k+1}}}\right)\sqrt{\log T}T^\rho + 12Kd\sqrt{\log T}T^\rho$$

$$\leq 144dK\sqrt{\log T}T^{\rho + \frac{3/2-\rho}{2^{k+1}}}.$$

The claim holds true for step $k+1$. Therefore, it implies that the claim holds true for all k. Repeating this process infinitely many times, we obtain

$$\text{Reg}_T \leq \mathcal{O}\left(d\sqrt{\log T}T^\rho\right),$$

which completes the proof when $\rho \in [0.5, 1]$. $\qquad\qquad\qquad\qquad\qquad\qquad\qquad\qquad\qquad\quad\square$

$\square$

# E  PROOF FOR PROPOSITION 2

**Proposition** (Proposition 2 restated). *If the total corruption level satisfies $\sum_{t=1}^{T} |c_t(a_t, a_t')| \leq \mathcal{O}(T^\rho)$, then for any $\alpha \in [\frac{1}{2}, 1)$, if the utility function is strongly concave, for sufficiently large round $T$, by choosing learning rate $\eta_\rho := \frac{\sqrt{\log T}}{2d} T^{-\alpha}$*

1. *(Robustness) RoSMID incurs $Reg_T \leq \tilde{\mathcal{O}}(dT^\alpha + \sqrt{d}T^{\frac{1}{2}(1-\alpha)+\rho} + dT^\rho)$.*

2. *(Efficiency) There exists a strongly concave utility function $\mu$ and a link function $\sigma$ such that $Reg_T$ suffered by RoSMID is at least $\Omega(T^\alpha)$ in scenario without corruption.*

## E.1  PROOF FOR ROBUSTNESS STATEMENT IN PROPOSITION 2

*Proof.* From Lemma 4.3, we know

$$\mathbb{E}\left\{\sum_{t=1}^{T} b_t^\top (a_t - a_T^*)\right\} \leq 2RdL^\sigma \sqrt{\lambda_R^* + 2\phi} T^\rho + d\sqrt{\lambda \eta_\rho} \mathbb{E}\left\{\sum_{t=1}^{T} \sqrt{t} \min\left(2, L^\sigma |c_t(a_t, a_t')|\right) \|a_t - a_T^*\|_2\right\}$$

$$\leq 2RdL^\sigma \sqrt{\lambda_R^* + 2\phi} T^\rho + 2Rd\sqrt{\lambda \eta_\rho} L^\sigma \sum_{t=1}^{T} \sqrt{t} |c_t(a_t, a_t')|$$

$$\leq 2RdL^\sigma \sqrt{\lambda_R^* + 2\phi} T^\rho + 2Rd\sqrt{\lambda \eta_\rho T} L^\sigma T^\rho.$$

We get the second inequality by using the fact that the diameter of the action space of $R$. We get the last inequality by using Abel Summation Equation (see Lemma D.7). Therefore, by Lemma 4.2, we have

$$\text{Reg}_T \leq \frac{C_0}{\eta_\rho} \log(T) + 8d^2 \eta_\rho T + 2L^\mu L_1^\sigma R + 4RdL^\sigma \sqrt{\lambda_R^* + 2\phi} T^\rho + 4Rd\sqrt{\lambda \eta_\rho T} L^\sigma T^\rho.$$

Choosing $\eta_\rho = \frac{\sqrt{\log T}}{2d} T^{-\alpha}, \alpha \in [0.5, 1)$, we have

$$\text{Reg}_T \leq 2dC_0 \sqrt{\log T} T^\alpha + 4d\sqrt{\log T} T^{1-\alpha} + 4RdL^\sigma \sqrt{\lambda_R^* + 2\phi} T^\rho + 4R\sqrt{d} L^\sigma (\log T)^{0.25} T^{\frac{1}{2}(1-\alpha)} T^\rho + 2L^\mu L_1^\sigma R$$

$$\leq O\left(d\sqrt{\log T} T^\alpha + \sqrt{d}(\log T)^{\frac{1}{4}} T^{\frac{1}{2}(1-\alpha)} T^\rho + dT^\rho\right),$$

which completes the proof of robustness statement. $\qquad\square$

## E.2  PROOF FOR EFFICIENCY STATEMENT IN PROPOSITION 2

*Proof.* The robustness statement in Proposition 2 implies that RoSMID could afford agnostic corruption level $\mathcal{O}(T^{\frac{1+\alpha}{2}})$. If Lemma E.1 holds, this implies that there exists a hard instance which makes RoSMID incur regret in order of $\Omega(T^\alpha)$ in non-corrupt setting. We can prove this by contradiction. Assume the efficiency statement in Proposition 2 is not true, which implies that for all $\mu$, and $\sigma$, there exists an $\epsilon > 0$ such that $\text{Reg}_T = c_0 T^{\alpha-\epsilon}$, for some constant $c_0 > 0$, we assume $\epsilon$ to be the least possible. In this scenario, when we consider a linear link function $\sigma(x) = \frac{1+x}{2}$, Lemma E.1 implies that an agnostic corruption budget $2\sqrt{c_0} T^{\frac{1+\alpha-\epsilon}{2}}$ suffices to induce linear regret, which contradicts the robustness statement in Proposition 2.

**Lemma E.1.** *For any algorithm $\mathcal{G}$ incurs regret $Reg_T \leq c_0 T^\alpha$, for some constant $c_0 > 0$, on learning from bandit with reward feedback for strongly concave utilities $\mu$ in non-corrupted setting, then there exists an instance $\mu$ such that $\mathcal{G}$ will suffer linear regret under agnostic corruption with budget $\sqrt{c_0} T^{\frac{1+\alpha}{2}}$.*

*Proof.* Consider the scenario when $d = 2$, the action space $\mathcal{A} := \{(a_1, a_2) : a_1 \geq 0, a_2 \geq 0, a_1 + a_2 \leq 1\}$, the utility function $\mu_\theta(a) = \langle \theta, a \rangle - \frac{1}{2}\|a\|_2^2$ with $\theta := [1, 0]$. In this scenario, the optimal action is $a^* := [1, 0]$. Assume that the adversary want to "fool" the algorithm $\mathcal{G}$ to make it believe the $\theta := [1, 1]$, with optimal arm $a^* = [\frac{1}{2}, \frac{1}{2}]$, each round, the corruption it should introduce is $c_t(a_t) = a_{t2}$, where $a_{ti}$ is the $i$-th coordinate value of action $a_t$. To make reward

indistinguishable from these two environments, the total corruption budget which the adversary has to pay is $\sum_{t=1}^{T} |a_{t2}|$. We know that $\mathcal{G}$ incurs regret $\text{Reg}_T := \sum_{t=1}^{T} \langle \theta, a^* - a_t \rangle - \frac{1}{2} \|a^*\|_2^2 + \frac{1}{2} \|a_t\|_2^2 = \frac{1}{2} - a_{t1} + \frac{1}{2} a_{t1}^2 + \frac{1}{2} a_{t2}^2 \leq c_0 T^\alpha$. Given this constraint, the maximum possible corruption paid by the adversary to make $\theta = [1, 1]$ and $\theta = [1, 0]$ indistinguishable is $\max \sum_{t=1}^{T} |a_{t2}| = \sqrt{c_0} T^{\frac{\alpha+1}{2}}$. Therefore, for any agnostic corruption no less than $\sqrt{c_0} T^{\frac{\alpha+1}{2}}$, $\mathcal{G}$ can not distinguish $\theta = [1, 0]$ and $\theta = [1, 1]$ and has to incur linear regret in either of these environments. □

□

## F    PROOF FOR PROPOSITION 3:

In this section, we present the regret upper for dueling bandit gradient descent Yue & Joachims (2009) in presence of *arbitrary* and *agnostic* corruption. Before presenting the proof, we make several remarks. First, in Yue & Joachims (2009), the utility function $\mu$ is assumed to be strictly concave to ensure the uniqueness of the maximizer. However, in scenarios where the maximizer is unique within the *action* set for a concave utility function $\mu$, the requirement for "strictness" can be relaxed. Second, we highlight the notation differences. We use $a$ to denote actions while Yue & Joachims (2009) uses $w$. In our proof, we define $P_t(a) := \sigma(\mu(a_t) - \mu(a))$ to represent the probability of the event $a_t \succ a$. While in Yue & Joachims (2009), it is denoted as $c_t(w)$. Just to highlight, its corrupted version $\hat{P}_t(a) := \sigma(\mu(a_t) - \mu(a) + c_t(a_t, a))$ is newly introduced in our paper. Moreover, $\bar{P}_t(a) := \mathbb{E}_{x \in \mathbb{B}} [P_t(\mathcal{P}_\mathcal{A}(a + \delta x))]$ is denoted as $\hat{\epsilon}_t(w) + \frac{1}{2}$ in Yue & Joachims (2009). The definition of $\mathcal{F}(\mathcal{P}_\mathcal{A}(a_t + \delta u_t), a_t)$ is same as $X_t(\mathcal{P}_W(w_t + \delta u_t))$ defined in Yue & Joachims (2009), except different notation. We remark $\mathcal{P}_\mathcal{A}(a)$ represent the projection of action $a$ on $\mathcal{A}$.

---

**Algorithm 3:** Dueling Bandit Gradient Descent (DBGD) (Yue & Joachims, 2009)

**Input:** Exploration rate $\delta$, exploitation rate $\gamma$, initial action $a_1 = \mathbf{0} \in \mathcal{A}$.

1  **for** $t \in [T]$ **do**
2  $\quad$ Sample unit vector $u_t$ uniformly and set $a_t' = \mathcal{P}_\mathcal{A}(a_t + \delta u_t)$.
3  $\quad$ Present action pair $(a_t, a_t')$ to user and receive corrupted dueling feedback $\hat{\mathcal{F}}(a_t', a_t)$.
4  $\quad$ Compute gradient $\hat{g}_t = -\frac{d}{\delta} \hat{\mathcal{F}}(a_t', a_t) u_t$.
5  $\quad$ Set learning rate $\eta = \frac{\gamma \delta}{d}$ and update $a_{t+1} = \mathcal{P}_\mathcal{A}(a_t - \eta \hat{g}_t)$.

---

**Proposition** (Proposition 3 restated). *If the total corruption level satisfies $\sum_{t=1}^{T} |c_t(a_t, a_t')| \leq \mathcal{O}(T^\rho)$, then for any $\alpha \in (0, \frac{1}{4}]$, if utility function $\mu$ is generally concave, for a sufficiently large round $T$, choosing $\gamma := \frac{R}{\sqrt{T}}$ and $\delta := \frac{\sqrt{2Rd}}{\sqrt{13L^\sigma L^\mu T^\alpha}}$ for Algorithm 3*

1. *(Robustness) $\text{Reg}_T$ incurred by Algorithm 3 satisfies $\text{Reg}_T \leq \mathcal{O}(\sqrt{d} T^{1-\alpha} + \sqrt{d} T^{\alpha+\rho})$.*

2. *(Efficiency) There exists a linear utility function $\mu$ and a link function $\sigma$ such that $\text{Reg}_T$ suffered by Algorithm 3 is at least $\Omega(T^{1-\alpha})$ in scenario without corruption.*

### F.1    PROOF FOR ROBUSTNESS STATEMENT IN PROPOSITION 3

*Proof.* The proof of Theorem 3 builds on the proof of Yue & Joachims (2009), where we extend it to a setting in presence of agnostic corruption. The proof follows Step 1-3 introduced in Theorem 4.1. Similarly, in Step 1, we decompose bias in the gradient caused by corruption. Notice that without corruption, DBGD performs expected gradient ascent, where the gradient $g_t$ is defined as follows

$$g_t = -\frac{d}{\delta} \mathcal{F}(\mathcal{P}_\mathcal{A}(a_t + \delta u_t), a_t) u_t.$$

In the presence of adversarial corruption, the corrupted gradient is

$$\hat{g}_t = -\frac{d}{\delta} \hat{\mathcal{F}}(\mathcal{P}_\mathcal{A}(a_t + \delta u_t), a_t) u_t.$$

Therefore, we quantify the bias $\mathbb{E}(b_t | a_t)$ in the following lemma.

**Lemma F.1.** *Let $P_t(a)$ represent the likelihood of action $a_t$ being preferred over $a$, where $P_t(a) := \sigma(\mu(a_t) - \mu(a))$. Likewise, the corrupted version is defined as $\hat{P}_t(a) = \sigma(\mu(a_t) - \mu(a) + c_t(a_t, a))$. The smoothed version of $P_t$ over $\mathcal{A}$ is $\bar{P}_t(a) := \mathbb{E}_{x \in \mathbb{B}}[P_t(\mathcal{P}_{\mathcal{A}}(a + \delta x))]$. Let $a'_t = \mathcal{P}_{\mathcal{A}}(a_t + \delta u_t)$, where $u_t$ is uniformly sampled from $\mathbb{S}$ and $\hat{g}_t = -\frac{d}{\delta} \hat{\mathcal{F}}(a'_t, a_t) u_t$. Let $b_t := \frac{d}{\delta}\left(\hat{P}_t(a'_t) - P_t(a'_t)\right) u_t$. We have $\mathbb{E}(\hat{g}_t|a_t) = \mathbb{E}(\nabla \bar{P}_t(a_t)|a_t) - \mathbb{E}(b_t|a_t)$.*

*Proof.* **(Proof for Lemma F.1).** Since $a'_t := \mathcal{P}_{\mathcal{A}}(a_t + \delta u_t)$, we have

$$
\begin{aligned}
\mathbb{E}(\hat{g}_t|a_t) &= \mathbb{E}_{u_t}(\mathbb{E}(\hat{g}_t|a_t, u_t)) \\
&= -\frac{d}{\delta} \mathbb{E}_{u_t}\left(\mathbb{E}(\hat{\mathcal{F}}(a'_t, a_t) u_t | a_t, u_t)\right) \\
&= -\frac{d}{\delta} \mathbb{E}\left(\hat{P}_t(a'_t) u_t | a_t\right) \\
&= -\frac{d}{\delta} \mathbb{E}\left(\left[P_t(a'_t) + \hat{P}_t(a'_t) - P_t(a'_t)\right] u_t | a_t\right) \\
&= \nabla \mathbb{E}_{x \in \mathbb{B}}\left(P_t(\mathcal{P}_{\mathcal{A}}(a_t + \delta x))|a_t\right) - \frac{d}{\delta} \mathbb{E}\left[\left(\hat{P}_t(a'_t) - P_t(a'_t)\right) u_t | a_t\right] \quad (23) \\
&= \nabla \bar{P}_t(a_t) - \frac{d}{\delta} \mathbb{E}\left[\left(\hat{P}_t(a'_t) - P_t(a'_t)\right) u_t | a_t\right] \\
&= \mathbb{E}(g_t|a_t) - \frac{d}{\delta} \mathbb{E}\left[\left(\hat{P}_t(a'_t) - P_t(a'_t)\right) u_t | a_t\right]. \quad (24)
\end{aligned}
$$

We get the equation equation 23 by using Lemma F.2. We get equation equation 24 by using Lemma F.3.

**Lemma F.2** (Lemma 2 in Yue & Joachims (2009)). *Fix $\delta > 0$, over random unit vector $u$, we have*

$$
\mathbb{E}[P_t(\mathcal{P}_{\mathcal{A}}(a + \delta u))u] = \frac{\delta}{d} \nabla \bar{P}_t(a).
$$

**Lemma F.3** (Lemma 1 in Yue & Joachims (2009)). $\mathbb{E}_{\mathcal{F}, u}[\mathcal{F}(a'_t, a_t)u] = -\mathbb{E}_u[P_t(a'_t)u]$.

$\square$

In Step 2, we decompose regret under dueling bandits into two parts, regret of decision and feedback error.

**Lemma F.4** (Regret Decomposition for DBGD). *Define $\lambda := \frac{L^\sigma}{L^\sigma - \delta L^\mu L_2}$, $L_2$ is the Lipschitz constant for $\sigma'$, $\gamma = R/\sqrt{T}$, for $b_t$ defined in Lemma F.1, we have*

$$
Reg_T \le \lambda \underbrace{\left(\frac{2Rd\sqrt{T}}{\delta} + 13\delta L^\sigma L^\mu T\right)}_{\textcolor{red}{Regret\ of\ Decision}} + 2\lambda \underbrace{\sum_{t=1}^{T} \mathbb{E}\left(b_t^\top (a_t - a^*)\right)}_{\textcolor{red}{Feedback\ Error}}.
$$

*Proof.* Because of Lemma F.5, we have

$$
\begin{aligned}
\mathbb{E}\left[\sum_{t=1}^{T} \bar{P}_t(a_t) - \bar{P}_t(a^*)\right] &\le \sum_{t=1}^{T} \mathbb{E}\left(\lambda \nabla \bar{P}_t(a_t)(a_t - a^*) + (3 + \lambda)\delta L^\sigma L^\mu\right) \\
&= \lambda \sum_{t=1}^{T} \mathbb{E}(\mathbb{E}(g_t|a_t)^\top (a_t - a^*)) + (3 + \lambda)\delta L^\sigma L^\mu T \\
&= \lambda \sum_{t=1}^{T} \mathbb{E}(\mathbb{E}(\hat{g}_t + b_t|a_t)^\top (a_t - a^*)) + (3 + \lambda)\delta L^\sigma L^\mu T \\
&= \lambda \sum_{t=1}^{T} \mathbb{E}(\hat{g}_t^\top (a_t - a^*)) + \lambda \sum_{t=1}^{T} \mathbb{E}(b_t^\top (a_t - a^*)) + (3 + \lambda)\delta L^\sigma L^\mu T
\end{aligned}
$$

**Lemma F.5** (Lemma 4 in Yue & Joachims (2009)). *Fix $\delta \in (0, \frac{L^\sigma}{L^\mu L_2})$, for $\lambda$ defined in Lemma F.4, we have*

$$\mathbb{E}\left[\sum_{t=1}^T \bar{P}_t(a_t) - \bar{P}_t(a^*)\right] \le \sum_{t=1}^T \mathbb{E}\left(\lambda \nabla \bar{P}_t(a_t)^\top (a_t - a^*) + (3 + \lambda)\delta L^\sigma L^\mu\right).$$

Since $a_{t+1} = \mathcal{P}_\mathcal{A}(a_t - \eta \hat{g}_t)$, and $\eta = \frac{\gamma\delta}{d}$, by applying the telescoping sum and because $a_1 = 0$, $\|g_t\|_2 \le G, \forall t$, we have

$$\lambda \sum_{t=1}^T \mathbb{E}\left(\hat{g}_t^\top (a_t - a^*)\right) \le \lambda \left(\frac{R^2}{2\eta} + \frac{T\eta G^2}{2}\right).$$

$\square$

Noticing that $\|b_t\|_2 \le \frac{d}{\delta} \min(2, L^\sigma |c_t(a_t, a'_t)|)$. Moreover, $\|a_t - a^*\|_2 \le 2R$. Therefore, in Step 3, we can control the feedback error by the following.

**Lemma F.6.** $\sum_{t=1}^T \mathbb{E}\left(b_t^\top (a_t - a^*)\right) \le 2RdL^\sigma T^\rho / \delta$.

*Proof.* **(Proof for Lemma F.6).**

$$\sum_{t=1}^T \mathbb{E}(b_t(a_t - a^*)) \le 2R\frac{d}{\delta} \sum_{t=1}^T \min(2, L^\sigma |c_t(a_t, a'_t)|)$$

$$\le 2R\frac{dL^\sigma}{\delta} \sum_{t=1}^T |c_t(a_t, a'_t)|$$

$$= 2R\frac{dL^\sigma T^\rho}{\delta}.$$

$\square$

**Lemma F.7** (Lemma 3 in Yue & Joachims (2009)). $\mathrm{Reg}_T \le 2\mathbb{E}\left[\sum_{t=1}^T \bar{P}_t(a_t) - \bar{P}_t(a^*)\right] + 5\delta L^\sigma L^\mu T$

Set $G = \frac{d}{\delta}$, and by Lemma F.7, we have

$$\mathrm{Reg}_T \le \lambda \left(\frac{2Rd\sqrt{T}}{\delta} + 13\delta L^\sigma L^\mu T\right) + 4R\frac{\lambda dL^\sigma T^\rho}{\delta}.$$

Therefore, by choosing $\delta := \frac{\sqrt{2Rd}}{\sqrt{13L^\sigma L^\mu}T^\alpha}, \gamma := \frac{R}{\sqrt{T}}$, and $T > \left(\frac{\sqrt{2Rd}L_\mu L_2}{\sqrt{13L^\sigma L^\mu}L^\sigma}\right)^4$, we have

$$\mathrm{Reg}_T \le 2\lambda_T \sqrt{26RdL^\sigma L^\mu}T^{1-\alpha} + 2L^\sigma \sqrt{26RdL^\sigma L^\mu}T^{\alpha+\rho}.$$

$\lambda_T = \frac{L^\sigma \sqrt{13L^\sigma L^\mu}T^\alpha}{L^\sigma \sqrt{13L^\sigma L^\mu}T^\alpha - L^\mu L_2\sqrt{2Rd}}$, completes the proof.

$\square$

## F.2 PROOF FOR EFFICIENCY STATEMENT IN PROPOSITION 3

*Proof.* The robustness statement in Proposition 3 implies that DBGD could afford agnostic corruption level $\mathcal{O}(T^{1-\alpha})$. This implies that there exists a hard instance which makes DBGD attain regret in order of $\Omega(T^{1-\alpha})$ in non-corrupt setting, formally described as follows. To start a proof by contradiction, we assume that the efficiency statement in Proposition 3 is false. Specifically, for all problem instance $\mu, \sigma$, there exist a $\epsilon > 0$, such that $\mathrm{Reg}_T \le \mathcal{O}(T^{1-\alpha-\epsilon})$. We assume $\epsilon$ to be the least possible. In particular, there exists a pair of instance $\mu, \sigma$ such that $\mathrm{Reg}_T = c_0 T^{1-\alpha-\epsilon}$, $c_0$ is a positive constant. In other words, it says that there exists a pair of instance $\mu, \sigma$ such that DBGD suffers regret in $\Theta(T^{1-\alpha-\epsilon})$.

Consider the following problem instance. $\mu(a) = \theta^\top a$, specifically $\mu$ is linear function. Let $d = 2$ with action set $\mathcal{A}_1 := \{(a_1, a_2) : a_1 \geq 0, a_2 \geq 0, \frac{1}{2}a_1 + a_2 - \frac{1}{4} \leq 0\}$ with $\theta = [\frac{1}{2}, \frac{1}{2}]$. The optimal arm is $a_1 = [\frac{1}{2}, 0]$, which is unique. Let the link function $\sigma(x) = \frac{1}{2} + \frac{1}{2}x$. It is easy to verify that it is rotational symmetric and Lipschitz. Denote $\text{Reg}_T$ as the regret occurred by DBGD on this problem instance. We know $\text{Reg}_T \leq \mathcal{O}(T^{1-\alpha-\epsilon})$. Consequently, it implies that $a_2$ is at most proposed $8c_0 T^{1-\alpha-\epsilon}$ times. This is because $\text{Reg}_T \leq c_0 T^{1-\alpha-\epsilon}$ and if the proposed action pair $(a_t, a_t')$ including $a_2$ at round $t$, it occurs regret at least $\frac{1}{8}$.

Now consider a different problem instance with the same $\mu, \sigma$ but different action set, which is $\mathcal{A}_2 := \{(a_1, a_2) : a_1 \geq 0, a_2 \geq 0, \frac{3}{2}a_1 + a_2 - \frac{3}{4} \leq 0\}$. The optimal arm in this action set $a_2 = [0, \frac{3}{4}]$, which is also unique. Proposing arm $a_1 = [\frac{1}{2}, 0]$ occurs at least $\frac{1}{8}$ regret. Consider an adversary that pulls the utility of $a_2$ from $\frac{3}{8}$ down to $\frac{1}{8}$ whenever it is pulled at a cost of $c_t(A_t) = \frac{1}{4}$. It is equivalent to say that every time when the proposed arm $w$ falls in the region $\mathcal{W} = \{a_1 \geq 0, a_2 \geq 0, \frac{1}{2}a_1 + a_2 - \frac{1}{4} \geq 0, \frac{3}{2}a_1 + a_2 - \frac{3}{4} \leq 0\}$, we assume that the utility is sampled from $\theta^\top \mathcal{P}_{\mathcal{A}_1}(w)$ and $|c_t(w)| = |\theta^\top \mathcal{P}_{\mathcal{A}_1}(w) - \theta^\top w| \leq \frac{1}{4}$. Choosing the total corruption budget as $2c_0 T^{1-\alpha-\epsilon}$, the adversary can afford to corrupt $8c_0 T^{1-\alpha-\epsilon}$ times.

This means that as long as the adversary is corrupting, the utility observed in the first problem instance is exactly the same as the utility observed in the second problem instance, in which we pull $a_2$ as most $8c_0 T^{1-\alpha-\epsilon}$. However, $a_2$ is the optimal arm in the second problem instance, which implies that the regret occurred on the second problem instance is at least $T - 8c_0 T^{1-\alpha-\epsilon}$, which is linear in $T$. This contradicts the robustness statement in Proposition 3, which says that when agnostic corruption is $2c_0 T^{1-\alpha-\epsilon}$, the regret upper bound is $O(T^{1-\epsilon})$, which is sublinear. To reconcile the conflicts, we must have the efficiency statement in Proposition 3 true to make the agnostic corruption budget at least $\Omega(T^{1-\alpha})$ to make the parallel world argument valid. □

# G  ADDITIONAL EXPERIMENTS

In this section, we list all the experiment details. At the beginning, we introduce the regret order fitting method, and baseline algorithms for comparison.

**Fitted Order of Regret.** We introduce the methodology that we use to compute the order of $\text{Reg}_T$ in terms of the number of iteration, $T$. To fit the order of the regret, we first convert it into $\log$ scale. Then we input the last $1\%$ of data, run linear regression, and use ordinary least squares to estimate the slope of the line, which is the *fitted order* of $\text{Reg}_T$.

**Baseline Algorithms.** We consider three baseline algorithms for comparison, Doubler (Ailon et al., 2014) and Sparring (Ailon et al., 2014)

**Doubler** is the first approach that transforms a dueling bandit problem into a standard multi-armed bandit (MAB) problem. It operates in epochs of exponentially increasing length: in each epoch, the left arm is sampled from a fixed distribution, and the right arm is selected using a MAB algorithm to minimize regret against the left arm. The feedback received by the MAB algorithm is the number of wins the right arm achieves compared to the left arm. Under linear link assumption, Doubler has been proven to experience regret as the same order as underlying MAB algorithm. For continuous *action* space and general concave utility, we choose Bandit Gradient Descent (BGD, (Flaxman et al., 2004)), with regret $\mathcal{O}(T^{3/4})$, as the underlying MAB algorithm.

**Sparring** initializes two MAB instances and lets them compete against each other. It is a heuristic improvement over Doubler. Although it does not come with a regret upper bound guarantee, it is reported to enjoy better performance compared to Doubler (Ailon et al., 2014). We also choose BGD as the underlying MAB algorithm.

To validate efficiency-robustness trade-off in Proposition 2 and Proposition 3, we try different $\alpha$ values. In the first row of Figure 3, we consider $\alpha = 0.05$ for DBGD and $\alpha = 0.9$ for RoSMID. We consider $\rho \in [0.5, 0.95]$. According to the theoretical prediction, both DBGD and RoSMID can tolerate at most $\mathcal{O}(T^{0.95})$ agnostic corruption, which aligns with experiment results. This is because the estimated order of regret is less than 1. In the second row of Figure 3, we consider $\alpha = 0.1$ for DBGD and $\alpha = 0.8$ for RoSMID. According to the theoretical prediction, both DBGD and RoSMID can tolerate at most $\mathcal{O}(T^{0.9})$ agnostic corruption. From the figure, we can see they

have smaller fitted order of regret when $\rho = 0.5$ while at a cost of tolerating smaller magnitude of agnostic corruption (it has linear regret when $\rho = 0.95$), which reveals intrinsic tradeoff between efficiency and robustness.

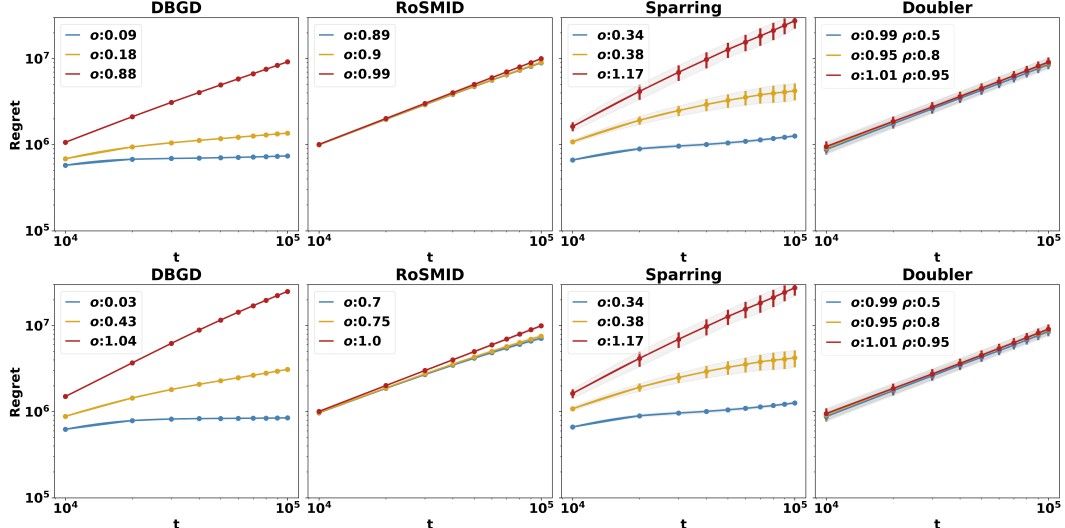

Figure 3: In the first row, we consider $\alpha = 0.05$ for DBGD and $\alpha = 0.9$ for RoSMID. In the second row, we consider $\alpha = 0.1$ for DBGD and $\alpha = 0.8$ for RoSMID. Given the $\alpha$, for each algorithm, we tested its performance under $\rho$-Imperfect Human feedback with $\rho = 0.5, 0.8, 0.95$. For each $\rho$, we presented a line plot of the average regret over five simulations, accompanied by $\pm$ one standard deviation shown by the shaded region. In the legend, $o$ denotes the estimated line slope, calculated using least squares on the last $1\%$ of the data.

### G.1 COMPUTATIONAL COMPLEXITY

Firstly, we emphasize that our robustified algorithm retains the same computational complexity as the originals. This indicates that incorporating robustness does not compromise computational efficiency, ensuring the practicality and scalability of the enhanced algorithm for real-world applications. Specifically, RoSMID has a running time of $O(d^3 T)$, where the $d^3$ term arises from the singular value decomposition. Similarly, the robustified DBGD algorithm runs in $O(dT)$, assuming that $P_A(a_t + \delta u_t)$ can be efficiently computed through matrix transformations.

As our work is the first to address corruption-robust learning from a continuous-action strictly concave utility, there is no directly comparable literature for this exact setting. However, we provide relevant comparisons for reference. RCDB (Di et al., 2024), designed for corruption-robust learning in a linear contextual dueling bandit framework, has a running time of $O(K^2 d^2 T)$, where $K$ is the number of actions. Similarly, Versatile-DB (Saha & Gaillard, 2022), which addresses dueling bandit problems in a multi-armed bandit setting under adversarial attacks, has a running time of $O(K^2 T)$, where $K$ represents the number of arms. $K^2$ comes from solving $\arg\max_{\boldsymbol{x} \in S}(\langle \boldsymbol{x}, \boldsymbol{y} \rangle - f(\boldsymbol{x}))$ using second order Newton methods and $f$ is a convex function.

