# OpenReview forum: "Learning from Imperfect  Human Feedback: A Tale from Corruption-Robust Dueling"
_ICLR.cc/2025/Conference — ICLR 2025 Poster_

### Official Review · Reviewer_5vny · 2024-10-25

**Soundness:** 3
**Presentation:** 3
**Contribution:** 3
**Rating:** 8
**Confidence:** 4

**Summary:**

This paper studies the problem of corrupted dueling bandits with strictly concave reward functions. It also considers a specific assumption (motivated by empirical observations on human behaviors), that the corruption budget at each time step decreases as $t^{\rho-1}$ for $\rho\in(0,1)$.

For that problem, the authors propose a lower bound in $\Omega(\max(\sqrt{T},T^{\rho}))$ and a stochastic mirror descent algorithm matching this bound, up to logarithmic terms.

**Strengths:**

It is the first work treating the problem of bandits with (strictly) concave utility functions and continuous actions spaces and it manages to do so with a quite complete picture and nice results :
- a lower bound, similar to linear bandits with corruption is proven: its proof nicely adapts to the fact that the corruption budget is individually bounded at each time step, rather than only having a bounded sum
- an algorithm with a nearly optimal (up to logarithmic terms) bound is proven. Moreover, its proof nicely constructs on the assumption that the corruption budget decay with time, making this assumption necessary for its good functioning

Overall, I find the problem interesting and the paper well written: it nicely discusses all the limitations of their results and why extending them is not obvious

**Weaknesses:**

My main concern is on the assumption of the link function. More precisely, it seems to me that the only link function $\sigma$ satisfying the conditions given between lines 143 and 153 is the constant function $1/2$.
I guess the reason for this is mainly due to the fact that either (or both) $\sigma$ should not be defined on $\mathbb{R}$ (but on an interval), nor that it is concave.

For instance, I would have liked that these conditions include the simoid function (which is the mostly used link function in the literature) and would suggest the authors to provide at least one example of link function that satisfies their conditions. Actually, I am quite surprised of the concavity assumption of $\sigma$. I guess a monotonicity assumption would be more desirable and natural here.

Additionally, this is not explicitly written, but it seems that the authors rely on the fact that $l_1^{\sigma}>0$ in the whole paper.

-------

Here are some minor remarks:
- I think the quantifiers in the lower bound results (Theorem 3.1, Lemma 3.1, Proposition) are given in a wrong order. It should typically be instead "for any learner, there exists an instance ..."

**Questions:**

- Can you relax the assumptions on the link function $\sigma$, so that it includes natural functions?
- Your algorithm requires a prior knowledge of the corruption budget $\rho$. Do you have any idea if we could design an algorithm with reasonable regret guarantees in the scenario where $\rho$ is unknown?

---

> ### Author Response · Authors · 2024-11-20
> **Response to Reviewer 5vny**
>
> We are happy to hear that the reviewer finds our paper “with a quite complete picture and nice results”. The reviewer has two major concerns/questions, which we believe we can fully address below (including clarifying a major misunderstanding about our assumption). Given the reviewer’s already positive impression and our resolution of both concerns, we would be grateful if the reviewer could consider re-evaluating our paper’s rating.
>
> **[W1 & Q1: stringent assumption on link functions]**
>
> We believe there might be some major misunderstandings here. Specifically, many natural link functions satisfy our assumptions, including the sigmoid function (as the reviewer mentioned), the cumulative Gaussian distribution function, and the linear function. We guess what the reviewer might have missed is that, we only require $\sigma(x)$ to be concave for x>0 (whereas the reviewer probably overlooked the x>0 condition and thought it has to be concave across the entire $\mathbb{R}$). The rotation-symmetry property then immediately implies that $\sigma(x)$ is convex for x<0. So it is really the same shape as sigmoid function (i.e., first convex and then concave), but is strictly more general by relaxing the specific format of sigmoid to a few smoothness properties. We remark that these assumptions about the link function are quite standard, and essentially no difference from the original (and widely-followed) dueling bandit paper in [Yue & Joachims, 2009; Kumagai, 2017].
>
> In fact, our purpose of imposing these assumptions is exactly trying to be more general than merely sigmoid link functions [Maystre & Grossglauser, 2017; Saha, 2021; Xu et al., 2024] or linear link function [Ailon et al., 2014; Chen & Frazier, 2017; Zimmert & Seldin, 2018; Lin & Lu, 2018], as assumed in some earlier works . We will revise our discussions here to more clearly reflect this as a strength of our work, rather than a limitation. **Please refer to the updated draft (lines 153-157)**.
>
>
> **[Q2: Regret upper bound for unknown $\rho$]**
>
> Please refer to the Common Response CR1, which we believe should fully address this question. We are happy to correspond with the reviewer if any further questions come up.
>
>
> **[Additional remarks]**
>
> Thank you for pointing out the inaccuracy in Lemma 3.1. In addition, we rely on the fact that $l^{\sigma}_1 > 0$. Since $R^{\mu}$ is finite, there exists a positive constants $l^{\sigma}_1 > 0$ and $L^{\sigma}_1 > 0$ such that $l^{\sigma}_1 \leq \sigma' \leq L^{\sigma}_1$ on $[- R^{\mu}, R^{\mu}]$. We made it explicit in the revised version. **Please refer to the updated draft (lines 153-157) and Lemma 3.1**.

---

> > ### Comment · Reviewer_5vny · 2024-11-25
> >
> > Thank you for your answer.
> >
> > You are right, I missed the fact that concavity was only needed on $\mathbb{R}_+$. As a minor comment, I would then suggest writing it as "concave on $\mathbb{R}^+$", as seeing concavity in a pointwise way seems weird to me.
> >
> > This was my main concern, which the authors clearly answered. After reading the other reviews, I thus decided to raise my score.

---

> > > ### Author Response · Authors · 2024-11-27
> > > **Response to Reviewer 5vny**
> > >
> > > We sincerely appreciate the reviewer’s very positive feedback. Based on the reviewer’s suggestion, we have updated the wording to "concave on $\mathbb{R}^{+}$" (highlighted in red at line 150). We look forward to engaging in further discussions with the reviewers in the next stage.

---

### Official Review · Reviewer_pdpN · 2024-11-03

**Soundness:** 3
**Presentation:** 3
**Contribution:** 3
**Rating:** 6
**Confidence:** 4

**Summary:**

This paper studies a new problem of LIHF under the dueling bandit setting with a continuous action space. This paper proposed a regret lower bound for this problem and then introduced a new algorithm based on gradient-based mirror descent that attains this lower bound. Experimental results are further conducted to validate the high efficiency of the proposed algorithm.

**Strengths:**

1. I didn't check all the proof in Appendix due to limited bandwidth, but after taking a close look at the theoretical results and the proof outlines, I feel the technical results are sound and reasonable.
2. The paper is clearly written and easy to understand.
3. This paper studies a timely topic of robust bandits to adversarial corruptions.
4. Experimental results are presented after theoretical analysis.

**Weaknesses:**

1. I feel there should be more literature review to help readers better understand your problem setting. I am not familiar with robust bandits, so after reading some existing literature, I suggest the authors to add the following discussion:

1.1 It's better to highlight the starting point of this line of work (Lykouris et al. Stochastic bandits robust to adversarial corruptions; Gupta et al. Better Algorithms for Stochastic Bandits with Adversarial Corruptions, etc.) and their formulation of this problem.

1.2 Since this paper consider the continuous action space/Lipschitz concave utility function under adversarial corruptions, it is better to mention some similar existing work on continuous action space such as robust kernelized bandit (Bogunovic et al. A robust phased elimination algorithm for corruption-tolerant Gaussian process bandits, etc.) and robust Lipschitz bandit (Kang et al. Robust Lipschitz bandit to adversarial corruptions, etc.), and discuss the connections with these existing works. It seems that this line of work still uses the classic bandit theory framework which is different from your gradient-based one, but I feel this point should be highlighted in your main paper.

1.3 More existing literature on gradient-based mirror descent should also be presented. It is important to let readers know how you modify the gradient-based idea to deal with the adversarial corruptions.

1.4 For the assumptions made in line 143-152, it is better to highlight where these assumptions come from one by one.

2. Since you studied a very new problem of decaying corruptions and argue that this is a practical problem, I feel it is better to add some real-world experimental results in your study. Simulations can hardly validate the high efficiency of your problem setting.

**Questions:**

1. Why do you think the assumptions made in line 143-152 are reasonable? I feel they are very specific.

2. Can you report the running time of your proposed algorithm compared with baselines? Since you study a practical problem and hence the algorithmic efficiency is also an important factor.

---

> ### Author Response · Authors · 2024-11-20
> **Response to Reviewer pdpN**
>
> **[W1.1 - 1.3:  discuss more relevant work]**
>
> We sincerely appreciate the reviewer's suggestion to include additional literature review, particularly regarding points 1.1, 1.2, and 1.3. One key distinction of our work lies in addressing corrupted dueling feedback, where the feedback is binary (0 or 1), providing significantly sparser information compared to standard bandit reward feedback, which contains observable numerical values. This fundamental difference partially explains why our methodology diverges from the approaches discussed in the suggested references.
>
> That said, we acknowledge the importance of providing a more comprehensive discussion and are willing to expand on these points. However, given the space constraints, we decide to prioritize comparing our methodology to more closely related work, as we mentioned in Comparisons with Related Works in the Introduction section.
>
>
> **[W1.4 & Q1: validity of assumptions]**
>
> We respectfully point out that while these assumptions appear “complex”, but at the heart they are mostly standard regularity and smoothness assumptions that have been widely used in previous literature as well. We provide detailed clarifications below.
>
> Assumption on link function:
>
> Our assumption already includes many natural link functions, such as sigmoid function, the cumulative Gaussian distribution function, and the linear function. In fact, our purpose of imposing these assumptions is exactly trying to be more general than merely sigmoid link functions [Maystre & Grossglauser, 2017; Saha, 2021; Xu et al., 2024] or linear link function [Ailon et al., 2014; Chen & Frazier, 2017; Zimmert & Seldin, 2018; Lin & Lu, 2018], as assumed in some earlier works . We will revise our discussions here to more clearly reflect this as a strength of our work, rather than a limitation. Please refer to the updated draft which highlighted in blue.
>
> Assumption on utility function:
>
> We assumed strong concave utility functions. This is standard due to at least two reasons. First, there is already a rich literature studying online optimization of strongly convex or concave functions, and our work subscribes to that rich literature [Shamir, 2013; Kumagai, 2017; Wan et al, 2021; Saha et al, 2022]. Second, strongly concave utilities  functions capture essential properties like diminishing marginal returns and risk-averse behavior. These characteristics align closely with real-world scenarios in economics, behavioral studies, and decision-making processes. Strongly concave functions are frequently utilized in economic literature for modeling utility functions such as the "indirect utility function" [Montrucchio, 1987; Sorger, 1995; Venditti, 1997], which represents the maximum utility a consumer can achieve based on a specific income level and set of prices.
>
> When the user exhibits strictly concave utility, we have theoretical guarantees in Proposition 3.
>
> Assumption on action spaces:
>
> The assumptions of a continuous action space are standard in the online convex optimization literature [Flaxman et al., 2004; Shamir, 2013; Kumagai, 2017;  Yue & Joachims, 2009]. Moreover, they are particularly natural for today’s preference learning for recommendation systems since languages, texts, videos are mostly processed as embedding vectors which are continuous by nature. The reviewer can find this motivation description in Line 93-96.
>
> **[Q2: Computational complexity and running time]**
> Please refer to the Common Response CR2 for computational complexity. For running time, for RoSMID, it takes around 45 minutes for running T= 10^5 with $d = 5$, and for DBGD, it takes around 21 minutes, using Google Colab.

---

> > ### Author Response · Authors · 2024-11-20
> > **References**
> >
> > Alain Venditti. Strong concavity properties of indirect utility functions in multisector optimal growth models. Journal of Economic Theory, 74(2):349–367, 1997.
> >
> > Gerhard Sorger. On the sensitivity of optimal growth paths. Journal of Mathematical Economics,24(4):353–369, 1995.
> >
> > Luigi Montrucchio. Lipschitz continuous policy functions for strongly concave optimization problems. Journal of mathematical economics, 16(3):259–273, 1987.
> >
> > Abraham D Flaxman, Adam Tauman Kalai, and H Brendan McMahan. Online convex optimization in the bandit setting: gradient descent without a gradient. arXiv preprint cs/0408007, 2004.
> >
> > Ohad Shamir. On the complexity of bandit and derivative-free stochastic convex optimization, 2013.
> >
> > Yue, Y., & Joachims, T. (2009, June). Interactively optimizing information retrieval systems as a dueling bandits problem. In Proceedings of the 26th Annual International Conference on Machine Learning (pp. 1201-1208).
> >
> > Wataru Kumagai. Regret analysis for continuous dueling bandit. Advances in Neural Information Processing Systems, 30, 2017.
> >
> > Wan, Y., Tu, W. W., & Zhang, L. (2022). Online strongly convex optimization with unknown delays. Machine Learning, 111(3), 871-893.
> >
> > Aadirupa Saha, Tomer Koren, and Yishay Mansour. Dueling convex optimization with general preferences, 2022. URL https://arxiv.org/abs/2210.02562.
> >
> > Lucas Maystre and Matthias Grossglauser. Just sort it! a simple and effective approach to active preference learning. In International Conference on Machine Learning, pp. 2344–2353. PMLR, 2017.
> >
> > Aadirupa Saha. Optimal algorithms for stochastic contextual preference bandits. Advances in Neural Information Processing Systems, 34:30050–30062, 2021.
> >
> > Nir Ailon, Thorsten Joachims, and Zohar Karnin. Reducing dueling bandits to cardinal bandits, 2014
> >
> > Bangrui Chen and Peter I Frazier. Dueling bandits with weak regret. In International Conference on Machine Learning, pp. 731–739. PMLR, 2017.
> >
> > Wenjie Xu, Wenbin Wang, Yuning Jiang, Bratislav Svetozarevic, and Colin N Jones. Principled preferential bayesian optimization. arXiv preprint arXiv:2402.05367, 2024.
> >
> > Julian Zimmert and Yevgeny Seldin. Factored bandits. Advances in Neural Information Processing Systems, 31, 2018.
> >
> > Chuang-Chieh Lin and Chi-Jen Lu. Efficient mechanisms for peer grading and dueling bandits. In Asian Conference on Machine Learning, pp. 740–755. PMLR, 2018.

---

> ### Comment · Reviewer_pdpN · 2024-11-20
> **Thanks for the response**
>
> I would like to appreciate authors for getting back to me on my concerns. I have a better understanding of your work now.
>
> For my concern 1.1-1.3, I understand that due to the space limit you can't include too much discussion to expand on these points. I know the dueling bandit only outputs binary feedback instead of a numeric one, but I can't tell if that will bring about extra difficulty. From my experience with dueling bandits, the technical analysis is not much more challenging and different compared with multi-armed ones. Therefore, I think it would be better to include more work on robust bandits and robust bandits on continuous space to highlight the technical contributions of yours. I kindly suggest adding a more comprehensive related work discussion as 1.1-1.3 at least in Appendix in the revision. That helps me understand your work a lot.
>
> For my concern 1.4 and Q1, I also recommend you add this detailed discussion in your Appendix along with the related work expansion in the revision. That also helps me appreciate your work, and otherwise these assumptions look not very natural to me
>
> For Q2, I think that is an important result and thanks for sharing. It would be better if you could add this running time information with more details in the revision, at least in Appendix.
>
> Thank you. I will still keep my rating as 6 on this work for now.

---

> > ### Author Response · Authors · 2024-11-22
> > **Response to Reviewer pdpN**
> >
> > We appreciate the reviewer’s kind suggestions. We will add more related work and discussion on computational complexities and validity of assumptions in the revised version.

---

> > > ### Comment · Reviewer_pdpN · 2024-11-25
> > >
> > > Thank you for your responses. I am unsure if the authors could make the revision in Appendix by the end of the discussion phase under the suggestions, but I will keep leaning on acceptance.

---

> > > > ### Author Response · Authors · 2024-11-27
> > > > **Response to Reviewer pdpN**
> > > >
> > > > We sincerely thank the reviewer for their patience as we worked to update the Appendix. We have added a detailed explanation in Appendix B regarding the challenges of developing robust algorithms for continuous dueling bandits with generally concave utility functions under corruption induced by strong adversaries. This section addresses all the literature mentioned in 1.1–1.3, along with additional relevant works.
> > > >
> > > > Furthermore, we have expanded our discussion on the generality of the assumptions in Appendix A and provided a detailed analysis of computational complexity in Appendix G.1. The main paper has also been updated accordingly. For your convenience, all updates are highlighted in red in the revised manuscript, which can be found in the revision file.

---

### Official Review · Reviewer_aLPp · 2024-11-08

**Soundness:** 3
**Presentation:** 2
**Contribution:** 3
**Rating:** 6
**Confidence:** 3

**Summary:**

This paper studies This paper studies Learning from Imperfect Human Feedback, where human's imperfection decays over time. The authors begin by establishing a theoretical lower bound, suggesting that learning from imperfect human feedback can be as challenging as learning under arbitrary adversarial corruption. They then propose a new algorithm RoSMID, designed for continuous action spaces, which achieves a near-optimal regret bound under decaying corruption.

**Strengths:**

- Learning from imperfect human feedback with decaying imperfection is interesting and well-motivated. This provides a valuable extension to traditional dueling bandit frameworks by allowing for dynamic, time-decaying corruption in feedback, aligning well with real-world applications.
- The authors establish a rigorous lower bound for regret under imperfect feedback, showing that even when corruption decays, learning remains as challenging as in adversarially corrupted settings. This theoretical insight highlights the inherent challenges in dealing with imperfect human feedback.
- RoSMID is a novel algorithm that achieves a near-optimal regret bound in the presence of decaying corruption. The algorithm's reliance on gradient-based techniques in the dueling bandit setting is an innovative approach that bridges traditional bandit learning with more advanced optimization methods.
- Extensive experiments validate the theoretical bounds. These experiments also underscore the efficiency-robustness trade-off, an important consideration for the settings where feedback corruption is inevitable.

**Weaknesses:**

- RoSMID requires prior knowledge of the decay rate $\rho$ of the human feedback, which may be an unrealistic assumption in practical settings. A discussion of this limitation and insights into possible relaxations would be beneficial.
- Though the theoretical result is new, it is not entirely surprising given the similarities between decaying corruption and adversarial corruption. A more detailed discussion on the difference between decaying corruption and adversarial corruption and how it informs the design of RoSMID would enhance the paper's contribution.
- While RoSMID's theoretical performance is strong, the empirical evaluation could be more comprehensive. For example, it would be useful for the authors to include insights into RoSMID's computational efficiency relative to other methods and its scalability with increasing problem size.

**Questions:**

- Could RoSMID use an upper bound on $\rho$ if the exact decay rate is unknown? How would this affect the performance of RoSMID?
- Is the continuous action space setting inherently more challenging to address than discrete action spaces under imperfect feedback? Could the proposed algorithm be adapted to discrete action spaces?

---

> ### Author Response · Authors · 2024-11-20
> **Response to Reviewer aLPp**
>
> We thank the reviewer for appreciating the value of our work. We respond to the reviewer’s criticisms mentioned in weaknesses and answer the raised questions in the following.
>
> **[W1 & Q1: knowledge of $\rho$]**
>
> In terms of Knowledge of $\rho$, please refer to the Common Response CR1.
>
> **[W2: discussion on how the design of RoSMID replies on decaying corruption assumption]**
>
> The design of RoSMID is not specifically motivated by the presence or absence of the decaying corruption assumption. Instead, the proof of the tight regret bound heavily relies on this assumption, as discussed extensively in Step 4 in Section 4. Without the decaying corruption assumption, it becomes theoretically challenging to derive a regret bound as tight as the one obtained under the decaying condition.
>
> This challenge arises because the effective bias term that contributes to the regret upper bound is given by $ c_t \cdot \sqrt{\lambda_{\max}(\nabla^2 R_t(a_t))} \cdot \|a_t - a^*\|_T $.
>
> Notably, $ \sqrt{\lambda_{\max}(\nabla^2 R_t(a_t))} $ scales as $ \sqrt{t} $. If $ c_t $ follows a decaying corruption condition, such as $c_t = t^{\rho-1} $, then in cases like $ \rho = \frac{1}{2} $, the product $ c_t \cdot \sqrt{\lambda_{\max}(\nabla^2 R_t(a_t))} $ stabilizes to a constant or increases monotonically when $ \rho > \frac{1}{2} $.
>
> This decaying property is theoretically advantageous because it introduces a well-behaved structure for the bias term. This structure can be effectively bounded using tools like the Abel summation lemma, facilitating the derivation of a tight regret bound.
>
> **[W3:  computational efficiency and scalability]**
>
> Please refer to the Common Response CR2.
>
> **[Q2: is the continuous action space setting inherently more challenging to address than discrete action spaces under imperfect feedback? Could the proposed algorithm be adapted to discrete action spaces]**
>
> It is an intriguing question whether learning in continuous action spaces is more challenging than in discrete action spaces, or vice versa. Currently, these two settings employ distinct methodologies: gradient-based methods are typically used for learning in continuous action spaces [Ailon et al., 2014; Yue & Joachims, 2009; Kumagai, 2017], whereas UCB-type algorithms are commonly applied in discrete action spaces for utility estimation [Saha, 2021, Di, et. al, 2024]. However, these approaches and the challenges they address are not directly comparable due to the inherent differences in the action spaces. Exploring whether a unified methodology can effectively handle both settings would be an intriguing and valuable direction for future research.
>
> **References**
>
> Nir Ailon, Thorsten Joachims, and Zohar Karnin. Reducing dueling bandits to cardinal bandits, 2014
>
> Yue, Y., & Joachims, T. (2009, June). Interactively optimizing information retrieval systems as a dueling bandits problem. In Proceedings of the 26th Annual International Conference on Machine Learning (pp. 1201-1208).
>
> Wataru Kumagai. Regret analysis for continuous dueling bandit. Advances in Neural Information Processing Systems, 30, 2017.
>
> Aadirupa Saha. Optimal algorithms for stochastic contextual preference bandits. Advances in Neural Information Processing Systems, 34:30050–30062, 2021.
>
> Qiwei Di, Jiafan He, and Quanquan Gu. Nearly optimal algorithms for contextual dueling bandits from adversarial feedback. arXiv preprint arXiv:2404.10776, 2024.

---

> > ### Comment · Reviewer_aLPp · 2024-11-21
> >
> > Thank you for your response. My concerns have been addressed in the rebuttal and I will maintain my score. I look forward to further discussions with the other reviewers in the next stage.

---

### Official Review · Reviewer_kiH5 · 2024-11-09

**Soundness:** 3
**Presentation:** 3
**Contribution:** 2
**Rating:** 6
**Confidence:** 4

**Summary:**

This paper studies the problem of dueling bandits with continuous actions when the outcomes are corrupted. In particular, the authors assume that the comparison at time is perturbed by an amount $c_t(a_t, a_t')$ which is $t^{\rho - 1}$. First, the authors demonstrate a lower bound of $\Omega(\max( \sqrt{T}, T^\rho ) )$ on the regret even when $\rho$ is known. The authors complement this result by proving a matching upper bound by robustifying stochastic mirror descent algorithms proposed in prior work on dueling bandits literature. Furthermore, they also study the setting where the total amount of corruption is bounded i.e. $\sum_t c_t(a_t, a_t') \le T^\rho$ and need not follow the decaying pattern. For this setting, robust versions of stochastic mirror descent and bandit gradient descent are studied.

**Strengths:**

1. The proposed algorithm (robust version of stochastic mirror descent) is simple and elegantly generalizes prior algorithm proposed in the literature on dueling bandits.

2. The paper is well-written and the authors have clearly explained the main difficulties in proving the regret upper bound. Moreover, it is also interesting to see that the regret lower and upper bounds match (up to logarithmic factors).

**Weaknesses:**

1. The model of corruption assumes that the amount of corruption decays over time. Therefore, when $t$ is large, the corruption encountered by the algorithm is quite small. I believe this is a strong assumption and severely restricts the adversary. Although the authors have results when the total amount of corruption is bounded (proposition 2 and 3) the bounds are suboptimal in such a setting.

2. Although the authors have motivated the use of decaying corruption, I believe human labelers often display persistence bias and therefore it is not clear that the corruption will decay to zero over time. Perhaps another setting could be that the feedbacks display some non-negligible bias over time.

3. The authors proposed several assumptions on the link function. For example, the derivatives are bounded (both below and above). This is not satisfied for many link functions e.g. logistic functions. Furthermore, the setting assumes that the utility function is strongly concave which is also a strong assumption.

4. The result is subsection 4.1 is presented as efficiency-robustness tradeoff in learning from imperfect feedback. However, the results are specific to two learning algorithms and it is not clear whether such a trade-off exists for any learning algorithm. In particular, can the authors provide an algorithm independent lower bound for this setting?

**Questions:**

1. Can you generalize your results to consider the case when the link function might have an unbounded derivative or non-smooth? This covers a large class of link functions.

2. What is the regret upper bound when the utility function is just concave (and not strongly concave)?

3. Finally, can you explain the weaker regret upper bounds for the setting of bounded corruption (subsection 4.1)? In particular, does there exist some algorithm that provides a matching bound (as stated in theorem 3.1), or is the lower bound even stronger in this case?

---

> ### Author Response · Authors · 2024-11-20
> **Response to Reviewer kiH5**
>
> We thank the reviewer for appreciating the value of our work, and believe most (if not all) of the reviewers’ concerns could be addressed. Please see our answers below and we would be eager to correspond with the reviewer, if there are any follow-up questions.
>
> **[W1 & W2: is the assumption of decaying corruption scales too strong?]**
>
> Our perspectives: it is a reasonable assumption for our purpose of learning from imperfect feedback. Certainly, coming up with better assumptions is an interesting future direction, but not a trivial task. The reviewer’s proposal of constant bias is also not ideal. Please see our more detailed clarifications below.
>
> First, we appreciate the reviewer’s suggestion that human labelers may exhibit persistence bias. Our model actually captures this situation as a boundary case, with $\rho = 1$ and total corruption linear in $T$. It is not difficult to realize that this setting is not learnable – suppose the reward is always corrupted by a small yet unknown constant c, an algorithm can never figure out this c hence necessarily suffers linear regret. We mention this to respectfully point out that coming up with more natural models may not be as easy as the reviewer had thought, and our model is a reasonable starting point which hopefully could spur more thoughts from the community.
>
> Second, the decaying constraint assumption may not be as restrictive as the reviewer might have thought. Particularly, we show that it does share the same learning lower bound as the scenario without such a constraint (see lines 229–240 and Theorem 3.1), implying that both scenarios exhibit comparable levels of learning difficulty.
>
> Finally, the decaying constraint assumption is indeed grounded in much human behavior studies, and we are not the first to adopt it in learning human preferences (see cited previous works such as  [Immorlica et al., 2020; Yao et al., 2022; Wang et al., 2023]). It simply captures the situations that noises or mistakes made by humans will decrease as time goes. While persistent bias may not always be like this, but many other human behaviors are like this. For instance, human labelers’ mistakes during labeling science questions will naturally decrease. Our methods are applicable to these situations.
>
> **References**
>
> Nicole Immorlica, Jieming Mao, Aleksandrs Slivkins, and Zhiwei Steven Wu. Incentivizing exploration with selective data disclosure. In Proceedings of the 21st ACM Conference on Economics and Computation, pp. 647–648, 2020.
>
> Chaoqi Wang, Ziyu Ye, Zhe Feng, Ashwinkumar Badanidiyuru, and Haifeng Xu. Follow-ups also matter: Improving contextual bandits via post-serving contexts, 2023.
>
> Fan Yao, Chuanhao Li, Denis Nekipelov, Hongning Wang, and Haifeng Xu. Learning from a learning user for optimal recommendations. In International Conference on Machine Learning, pp. 25382–25406. PMLR, 2022.

---

> > ### Author Response · Authors · 2024-11-20
> > **[W1: suboptimality of Proposition 2 & 3]**
> >
> > We acknowledge that the regret bounds provided in Propositions 2 and 3 might be suboptimal. However, as discussed at the beginning of Section 4.1, the scenarios addressed in Propositions 2 and 3 are more general and challenging due to two key aspects: first, they consider generally concave utility functions, and second, they account for arbitrary adversarial corruption. Theoretical regret lower and upper bounds remain largely unexplored in these settings. Despite this complexity, Propositions 2 and 3 represent the first attempt to establish theoretical guarantees for concave utilities under arbitrary adversarial corruption in continuous action spaces, contributing to the literature. In a more specific scenario, however, our results are optimal when the utility function is strongly concave, and the corruption exhibits a decaying pattern (see Theorem 4.1).

---

> > > ### Author Response · Authors · 2024-11-20
> > > **[W3 & Q1: assumption on the link function]**
> > >
> > > We believe this question might be due to some misunderstanding. Specifically, the derivative of the logistic function, i.e. $\frac{1}{1 + \exp(-\lambda x)}$, is always bounded for natural situations – it is only unbounded when $ \lambda $ goes to infinity, which very rarely happens (PS: no dueling bandit algorithms can work in this case anyway). Therefore, almost all logistic functions satisfy our assumption.
> > >
> > > While we appreciate the reviewer’s suggestion of considering link functions $\sigma(x)$ that exhibit unbounded derivatives and nonsmooth behavior, it is unclear whether this is a natural question. To the best of our knowledge, we are not aware of any previous works in dueling bandits that consider non-smooth link function with unbounded derivatives – we would very much appreciate it if the reviewer could point us to any related references on this

---

> ### Author Response · Authors · 2024-11-20
> **[W3 & Q2: on strongly concave utility assumption, and what do we know beyond strong concavity? ]**
>
> First, we argue that strong concavity is a standard assumption, due to at least two reasons. (A)  There are already a rich literature studying online optimization of strongly convex or concave functions, and our work subscribes to that rich literature [Shamir, 2013; Kumagai, 2017; Wan et al, 2021; Saha et al, 2022]. (B) strongly concave utilities functions capture essential properties like diminishing marginal returns and risk-averse behavior. These characteristics align closely with real-world scenarios in economics, behavioral studies, and decision-making processes. Strongly concave functions are frequently utilized in economic literature, particularly in modeling utility functions such as the "indirect utility function" [Montrucchio, 1987; Sorger, 1995; Venditti, 1997], which represents the maximum utility a consumer can achieve based on a specific income level and set of prices.
>
> Second, we do have results under weaker utility assumption for strictly concave utility, though we did not view that as our main results, and embedded into the robustness-efficiency tradeoff analysis since we thought that is a deeper and more insightful phenomenon. For example,  Proposition 3 extends the current state-of-the-art DBGD algorithm, which is capable of learning from generally concave utilities, by generalizing its regret bound of $O(\sqrt{d}T^{\frac{3}{4}})$ (achieved with $\alpha = \frac{1}{4}$) to a more flexible bound of $O(\sqrt{d}T^{1 - \alpha} + \sqrt{d}T^{\alpha + \rho})$, where the total corruption is bounded by $T^{\rho}$. This result highlights the trade-off between learning efficiency and adversarial robustness, which can be achieved by appropriately tuning the learning rate, as discussed in lines 464–469.
>
> **References**
>
> Alain Venditti. Strong concavity properties of indirect utility functions in multisector optimal growth models. Journal of Economic Theory, 74(2):349–367, 1997.
>
> Gerhard Sorger. On the sensitivity of optimal growth paths. Journal of Mathematical Economics,24(4):353–369, 1995.
>
> Luigi Montrucchio. Lipschitz continuous policy functions for strongly concave optimization problems. Journal of mathematical economics, 16(3):259–273, 1987.
>
> Ohad Shamir. On the complexity of bandit and derivative-free stochastic convex optimization, 2013.
>
> Aadirupa Saha, Tomer Koren, and Yishay Mansour. Dueling convex optimization with general preferences, 2022. URL https://arxiv.org/abs/2210.02562.
>
> Wan, Y., Tu, W. W., & Zhang, L. (2022). Online strongly convex optimization with unknown delays. Machine Learning, 111(3), 871-893.

---

> ### Author Response · Authors · 2024-11-20
> **[W4: algorithmic independent lower bounds]**
>
> We do have **algorithm-independent** lower bounds in this setting and the trade-off is in fact inherent. **Please refer to the following two algorithm-independent lower bounds and our updated draft lines 477-481**. The algorithm-independent lower bound are stated as follows.
>
> 1. For any learner with uncorrupted regret bound $\text{Reg}$, there exists a linear utility function $\mu$ and a link function $\sigma$ such that when the total corruption level $C = \Theta(\text{Reg})$, it will induce linear regret (See Appendix D.2).
>
> 2. For any learner with an uncorrupted regret bound $\text{Reg}$, there exists a strongly concave utility function $\mu$ and a link function $\sigma$ such that, when the total corruption level $C = \Theta(\sqrt{\text{Reg} \cdot T})$, the learner will incur linear regret (see Lemma C.1).
>
> These algorithm-independent lower bounds demonstrate that improved learning efficiency comes at the expense of reduced robustness against unknown adversarial corruption.

---

> ### Author Response · Authors · 2024-11-20
> **[Q3: how to understand weaker regret upper bound]**
>
> We acknowledge that it indeed remains unclear whether an algorithm exists that achieves a matching bound as stated in Theorem 3.1 or if the lower bound for learning strongly concave utility functions under arbitrary adversarial corruption is inherently stronger. Closing this gap turns out to be a quite non-trivial question, despite our continuous effort even after the deadline. Currently, our gathered results and insights seem to support the conjecture that such an algorithm may be  achievable. We have also mentioned in lines 536–538 that an intriguing direction for future research is to develop algorithms that achieve $\mathcal{O}(\sqrt{T} + T^{\rho})$ regret for strongly concave utilities under arbitrary adversarial corruption.
>
> Furthermore, learning from strictly concave utility functions remains a longstanding open problem that has persisted for approximately 20 years [Belmega et al., 2018]. Specifically, there is a gap between the $ \Omega(\sqrt{T}) $ lower bound and the current best-known algorithms for learning strictly concave functions under bandit feedback, which only achieve a regret of $ O(T^{\frac{3}{4}}) $. Therefore, it is still meaningful to provide a robustness of the state-of-art algorithm under unknown corruption, even though the bound on $ T $ might not be optimal.
>
> **Reference**
>
> Belmega, E. V., Mertikopoulos, P., Negrel, R., & Sanguinetti, L. (2018). Online convex optimization and no-regret learning: Algorithms, guarantees and applications. arXiv preprint arXiv:1804.04529.

---

> > ### Comment · Reviewer_kiH5 · 2024-11-25
> > **Follow up**
> >
> > Dear Authors,
> >
> > Thanks for your detailed response. I agree that the trade-off result can be extended for algorithm-independent setting and the assumption on the link function is standard. Therefore, I have updated my score. However, I still believe that the assumption on decaying corruption is quite strong.

---

### Official Review · Reviewer_NYfx · 2024-11-11

**Soundness:** 3
**Presentation:** 3
**Contribution:** 3
**Rating:** 6
**Confidence:** 3

**Summary:**

This paper studied the dueling bandit problem with constraint corruption. It first proved that the problem is not easier with the constraint on the corruption with a lower bound. Next, it showed the efficiency of the proposed RoSMID algorithm with theoretical upper bound and numerical experiments. It also discussed the application of its analytical techniques.

==========

After rebuttal: I would like to keep the score.

**Strengths:**

1. The paper is generally easy to follow.
1. The proposed RoSMID algorithm is nearly optimal when the gap between its upper bound and the lower bound is only $\log T$.

**Weaknesses:**

1. The paper repeatedly emphasize the novelty of the analysis. It would be appreciated to summarize the novelty in a few words.
1. Lemma 3.1: I am confused why the author(s) discussed the direct reward feedback setting here.

**Questions:**

Please refer to the 'Weaknesses' section.

---

> ### Author Response · Authors · 2024-11-20
> **Response to Reviewer NYfx**
>
> **[W1: better emphasis of analysis novelty]**
>
> We thank the reviewer for their feedback. Our main technical contribution is a new framework for analyzing the regret of gradient-based algorithms under arbitrary corruptions. This framework introduces a novel regret decomposition lemma for dueling bandits, which upper bounds total regret by combining decision regret and feedback error due to corruption, based on quantifying bias in gradient estimation. **Please refer to our updated draft in the introduction section which now highlights this contribution in blue for your suggestions.**
>
>
> **[W2: why need a lemma (i.e., Lemma 3.1) to discuss direct reward feedback?]**
>
> This lemma is stated just as an intermediate step for the proof of Theorem 3.1, which is why it is named a lemma. At a high level, the proof of Theorem 3.1 has two major steps: (1) proving a lower bound for the direct reward feedback setting (intuitively an easier problem); (2) converting the above lower bound to the (intuitively more difficult) problem of dueling feedback. It just happened so that step (1) above was the core difficulty of the proof, hence we stated it as a lemma.

---

> > ### Comment · Reviewer_NYfx · 2024-11-25
> >
> > Thanks for your response. I would keep the current score.

---

### Author Response · Authors · 2024-11-20
**Common Response 1 (CR1): regret bounds for unknown $\rho$**

We appreciate the reviewers' overall positive evaluations about our work, especially the acknowledgement of the significance of the problem we study and the design of RoSMID. We are happy to integrate the reviewers’ suggestions for improving the current version.

In the following, we first answer two common questions raised by reviewers, and then address each reviewer’s questions separately.

Some reviewers have questions about whether our algorithm, RoSMID, can handle scenarios where $ \rho $ is unknown. The **quick answer is Yes** (which is precisely what our Section 4.1 is trying to address), but one has to suffer slightly worse regret bounds due to not knowing $\rho$ exactly (see Proposition 2 and Proposition 3).

Taking Proposition 2 as an example.  By choosing the learning rate of RoSMID $\eta = \frac{\sqrt{\log(T)}}{2dT^{\alpha}}$ for $\alpha \in [½, 1)$, it has regret $O(T^{\alpha} + T^{(1-\alpha)/2+\rho} + T^{\rho})$ when the total attack is $O(T^{\rho})$. Notably, the algorithm RoSMID here and its learning rate $\eta$ do not rely on knowledge of $\rho$ (**as desired by the reviewers**), but its performance depends on total amount of attacks, i.e., the $\rho$. Hence RoSMID can guarantee sublinear regret for any attack level $\rho < (1+\alpha)/2$, which is larger (i.e., more robust) when $\alpha$ increases. However, this comes with a cost since RoSMID’s regret under benign case without attack is $O(T^{\alpha})$, as stated in the second claim in Proposition 2, which increases (i.e., less efficient) as $\alpha$ increases. This tradeoff of the designed algorithm is precisely what we meant by  “efficiency-robustness tradeoff” (as discussed in lines 464-468), and is an interesting phenomenon that we wanted to highlight.

Proposition 3 offers a similar “efficiency-robustness tradeoff”, but for a different algorithm, i.e., the widely-adopted Dueling Bandit Gradient Descent (DBGD) algorithm. We present both results to showcase the usefulness of our novel analysis framework.

Meanwhile, we want to explicitly point out that the “efficiency-robustness tradeoff” is inherent under unknown adversarial attack. This fact is proved by the following **algorithm-independent lower bounds**.

1. For any learner with uncorrupted regret bound $\text{Reg}$, there exists a linear utility function $\mu$ and a link function $\sigma$ such that when the total corruption level $C = \Theta(\text{Reg})$, it will induce linear regret (See Appendix D.2).

2. For any learner with an uncorrupted regret bound $\text{Reg}$, there exists a strongly concave utility function \(\mu\) and a link function $\sigma$ such that, when the total corruption level $C = \Theta(\sqrt{\text{Reg} \cdot T})$, the learner will incur linear regret (see Lemma C.1).

Notably, we are not sure whether these bounds are tight (likely not), but they do suffice to demonstrate the fundamental message that improved learning efficiency has to come at the expense of reduced robustness against unknown adversarial corruption.

It is a very intriguing fundamental open question to study what is the **optimal tradeoff** under unknown adversarial corruption and how to achieve it. Our Proposition 2 and 3 serve as two examples of how we can trade off efficiency for robustness under two popular algorithm design paradigms.

---

> ### Author Response · Authors · 2024-11-20
> **Common Response 2 (CR2): computational complexity**
>
> Firstly, we would like to highlight that our robustified algorithm maintains the same computational complexity as the original algorithm of NC-SMD [Kumagai, 2017]. This demonstrates that our addition of robustness did not compromise computational efficiency, ensuring that the enhanced algorithm remains practical and scalable for real-world applications.
>
> RoSMID’s running time is $O(d^3 T)$, where d^3 comes from singular value decomposition.
>
> Robustified DBGD runs in $O(dT)$, if we assume $P_{A}(a_t + \delta u_t)$ can be effectively performed by matrix transformation
>
> Since we are the first to address corruption-robust learning from a continuous-action strictly-concave utility, there is no existing literature that directly matches this exact setting for comparison. However, we provide some closely relevant ones for the reviewers’ reference.
>
> 1. RCDB’s [Di. et al, 2024] running time is $O(K^2 d^2 T)$ for learning from linear contextual dueling bandit under corruption,  $K$ is the number of actions
>
> 2. Versatile-DB’s [Saha & Gaillard, 2022] running time is $O(K^2 T)$ for learning from dueling bandit in multi-armed setting under adversarial attacks, $K$ is the number of arms
>
> **References:**
>
> Qiwei Di, Jiafan He, and Quanquan Gu. Nearly optimal algorithms for contextual dueling bandits from adversarial feedback. arXiv preprint arXiv:2404.10776, 2024.
>
> Aadirupa Saha and Pierre Gaillard. Versatile dueling bandits: Best-of-both world analyses for learning from relative preferences. In Kamalika Chaudhuri, Stefanie Jegelka, Le Song, Csaba Szepesvari, Gang Niu, and Sivan Sabato (eds.), Proceedings of the 39th International Conference on Machine Learning, volume 162 of Proceedings of Machine Learning Research, pp. 19011–19026. PMLR, 17–23 Jul 2022. URL https://proceedings.mlr.press/v162/saha22a.html.

---

### Meta-Review · Area_Chair_fmXD · 2024-12-22

**Metareview:**

This paper addresses the concave-utility continuous-action dueling bandit problem, particularly focusing on a model that includes time-decaying corruption with an eye toward applications in Learning from Imperfect Human Feedback. The authors propose an algorithm for this problem setting and provide regret upper bounds. The effectiveness of the results is also validated through experiments.

Weaknesses of the paper include concerns that the scenario where corruption decays at a fixed rate is overly restrictive, that knowledge of the decay rate parameter $\rho$ is required in advance to achieve optimal bounds, and that the computational complexity is significant. However, the authors have provided reasonably convincing rebuttals to these concerns.

Most of the reviewers rated the paper near the borderline, but there was overall agreement on its positive contributions. As such, I support its acceptance.

**Additional Comments On Reviewer Discussion:**

Concerns have been raised regarding the restrictive assumption that corruption decays at a fixed rate, the requirement to know the decay parameter $\rho$ in advance to achieve optimal bounds, and the high computational complexity. Regarding the second point, the authors have shown that their approach can handle situations where $\rho$ is unknown, albeit with some degradation in the regret upper bound, and they have provided a reasonable interpretation and justification for this scenario.

While the assumption of strong corruption decay remains a valid concern and has not been fully addressed, the reviewers’ overall assessment of the paper is favorable, with no objections to acceptance. Therefore, I support its acceptance.

---

### Decision · Program_Chairs · 2025-01-22

Accept (Poster)